# Mega-TTS 2: Boosting Prompting Mechanisms for Zero-Shot Speech Synthesis

**Ziyue Jiang** [*†♠♡]   **Jinglin Liu** [*♡]   **Yi Ren** [*♡]   **Jinzheng He** [*†♠♡]   **Zhenhui Ye** [*†♠♡]
**Shengpeng Ji** [♠]   **Qian Yang** [♠]   **Chen Zhang** [♡]   **Pengfei Wei** [♡]   **Chunfeng Wang** [♡]
**Xiang Yin** [♡]   **Zejun Ma** [♡]   **Zhou Zhao** [♠‡]
[♠]Zhejiang University & [♡]ByteDance
{ziyuejiang,zhaozhou}@zju.edu.cn,
{liu.jinglin,ren.yi,yinxiang.stephen}@bytedance.com

## Abstract

Zero-shot text-to-speech (TTS) aims to synthesize voices with unseen speech prompts, which significantly reduces the data and computation requirements for voice cloning by skipping the fine-tuning process. However, the prompting mechanisms of zero-shot TTS still face challenges in the following aspects: 1) previous works of zero-shot TTS are typically trained with single-sentence prompts, which significantly restricts their performance when the data is relatively sufficient during the inference stage. 2) The prosodic information in prompts is highly coupled with timbre, making it untransferable to each other. This paper introduces Mega-TTS 2, a generic prompting mechanism for zero-shot TTS, to tackle the aforementioned challenges. Specifically, we design a powerful acoustic autoencoder that separately encodes the prosody and timbre information into the compressed latent space while providing high-quality reconstructions. Then, we propose a multi-reference timbre encoder and a prosody latent language model (P-LLM) to extract useful information from multi-sentence prompts. We further leverage the probabilities derived from multiple P-LLM outputs to produce transferable and controllable prosody. Experimental results demonstrate that Mega-TTS 2 could not only synthesize identity-preserving speech with a short prompt of an unseen speaker from arbitrary sources but consistently outperform the fine-tuning method when the volume of data ranges from 10 seconds to 5 minutes. Furthermore, our method enables to transfer various speaking styles to the target timbre in a fine-grained and controlled manner. Audio samples can be found in https://boostprompt.github.io/boostprompt/.

## 1 Introduction

In recent years, there has been remarkable progress in the development of text-to-speech (TTS) technology (Shen et al., 2018; Jia et al., 2018; Li et al., 2019; Kim et al., 2020; Ren et al., 2019; 2020; Kim et al., 2021; 2022a). Among them, adaptive TTS systems (Chen et al., 2021; Min et al., 2021; Kim et al., 2022b) are capable of cloning personalized voices given a few minutes of speech data. However, the performance of these systems relies heavily on the quality and quantity of the data utilized during the fine-tuning phases (Tan et al., 2021). Insufficient data during the fine-tuning stages can lead to diminished audio naturalness or speech intelligibility (Kang et al., 2023). Moreover, the computational demands also constrain its application for cloning everyone's voice.

To reduce such a reliance, existing works leverage generative models to perform zero-shot TTS (Cooper et al., 2020a; Casanova et al., 2022; Huang et al., 2022a; Kang et al., 2023; Kharitonov et al., 2023; Wang et al., 2023; Shen et al., 2023b; Matthew et al., 2023). These powerful models can effectively synthesize speech given only a single speech prompt, eliminating the need for data

---

[*]Equal contribution.

[†]Interns at ByteDance.

[‡]Corresponding author.

preparation and the computational requirements for fine-tuning methods. However, the prompting mechanisms of current solutions still face two primary challenges:

- **Lack of multi-sentence prompting strategies.** Previous works of zero-shot TTS typically employ single-sentence speech prompts during training (Wang et al., 2023; Shen et al., 2023b; Matthew et al., 2023). In inference, the information in the single-sentence speech prompt is insufficient to guide the zero-shot TTS systems to imitate the voice variability of a natural person perfectly.[1] From another perspective, the performance of fine-tuning methods can be further improved by increasing the amount of data, while zero-shot TTS systems lack an appropriate strategy to extract useful information from multi-sentence speech prompts.

- **Lack of specialized prompting mechanism for prosodic information.** Current solutions for zero-shot TTS primarily concentrate on improving the similarity of timbre and prosody between the generated speech and the prompts. However, they neglect to express various unseen prosodic styles in a controlled manner while also preserving the unique timbre of the given one-sentence prompt. In order to control the prosodic styles, it is necessary to disentangle the prosody information from speech prompts.

We address the above challenges by decomposing speech into content, timbre, and prosody. Intuitively, representing speeches for numerous speakers requires a substantial number of codebook entries for timbre modeling (Défossez et al., 2022; Yang et al., 2023). Through the decoupling of prosody information, a highly compact codebook for prosody modeling can be obtained, which enables our model to effectively handle extremely long prompts and have flexible control over prosodic styles. Therefore, this work proposes Mega-TTS 2, a generic framework that boosts the prompting mechanisms for zero-shot TTS systems. Specifically, we begin by designing an acoustic autoencoder that can effectively decompose speech into prosody and timbre representations and represent them in a compact latent space. Then, we design a multi-reference timbre encoder (MRTE) and a prosody latent language model (P-LLM) to extract useful information from multi-sentence prompts. In addition to the multi-sentence prompting mechanism, we propose a prosody interpolation technique to control the generation process of prosody codes by utilizing prosody prompts from multiple speakers while maintaining the target speaker's timbre. By utilizing the probabilities derived from both the prosodic prompts of the target speaker and the auxiliary speaker, the prosodic styles of speech can be generated in a controlled manner.

Experiments on LibriSpeech test-clean (Panayotov et al., 2015) and ESD (Zhou et al., 2021) datasets show that Mega-TTS 2 outperforms other state-of-the-art fine-tuning and zero-shot TTS models in terms of speaker similarity and speech naturalness. Notably, when the length of the prompt is further extended, our method surpasses the fine-tuning baseline model in the objective and subjective evaluations. The extensive studies on adaptive prosody transfer further highlight the superiority of our proposed prompting mechanisms. The main contributions of this work are summarized as follows:

- We design an acoustic autoencoder that separately compresses the prosody and timbre information into the latent space, which allows our model to process prompts of up to 300 seconds in length effectively.

- We propose a multi-reference timbre encoder and an auto-regressive prosody language model to extract fine-grained information from multiple reference speeches, which bridges the speaker similarity gap between zero-shot methods and fine-tuning methods.

- Experimental results also reveal that the performance of Mega-TTS 2 surpasses the powerful fine-tuning baseline when we have 10 seconds to 5 minutes of data for each unseen speaker, indicating the superiority of our proposed prompting mechanisms.

- The proposed prosody interpolation technique ensures the controllability of prosody and is capable of transferring various speaking styles to the desired timbre. For instance, we can transform a voice with a sad tone into a happier one with the auxiliary prosody prompt from another speaker.

---

[1]Although the performance of these systems can be further improved by concatenating multiple sentences into a long prompt, the gap between training and inference still restricts their performance. (See Section 4.2)

## 2 BACKGROUND

**Adaptive TTS**   Adaptive TTS (Arik et al., 2018; Kons et al., 2019; Moss et al., 2020; Chien et al., 2021) focuses on synthesizing personalized voice for any user with few data. During the adaptation process, a TTS model pre-trained on a multi-speaker speech dataset is typically fine-tuned with few adaptation data for the target voice (Tan et al., 2021). Chen et al. (2018) design independent learned embeddings for each speaker, which requires few data at deployment time to adapt to new speakers rapidly. AdaSpeech (Chen et al., 2021) proposes an acoustic-condition modeling method for high-quality and efficient customization of new voices. There are also some works leveraging the meta-learning approach (Chen et al., 2018; Min et al., 2021; Huang et al., 2022b) and data augmentation (Cooper et al., 2020b; Yang & He, 2020) for speaker adaptation. However, although some works are data-efficient (Min et al., 2021; Huang et al., 2022b) and parameter-efficient (Chen et al., 2021), these systems still suffer from audio quality issues when data size is small, as well as computational cost issues due to hundreds of fine-tuning steps.

**Zero-shot TTS**   Zero-shot adaptation (Jia et al., 2018; Arik et al., 2018; Cooper et al., 2020a; Casanova et al., 2021; Wu et al., 2022; Huang et al., 2022b;a; Casanova et al., 2022) aims to synthesize unseen voices with a speaker encoder that extracts speaker embeddings from the reference audio. This scenario is highly attractive because it does not require any adaptation data or parameters (Kang et al., 2022). The attention-based adaptation method (Choi et al., 2020; Zhou et al., 2022; Yin et al., 2022; Lin et al., 2021) utilizes attention mechanisms to extract fine-grained speech features from reference audios. Among them, Attentron (Choi et al., 2020) proposes to extracts useful style information from arbitrary number of reference audios. However, they do not separately model the timbre and prosody information, lacking controllability over timbre and prosody. Most recently, some works (Kharitonov et al., 2023; Zhang et al., 2023) are proposed to use in-context learning methods (Dong et al., 2022) to efficiently extract speaker information from acoustic prompts and have achieved remarkable results in zero-shot TTS. VALL-E (Wang et al., 2023) proposes the neural codec language model that exhibits strong in-context learning capability for zero-shot speech generation. NaturalSpeech 2 (Shen et al., 2023b) introduces in-context learning to latent diffusion model (Rombach et al., 2022), which is achieved by partitioning a speech clip into the prompt and target regions. VoiceBox (Matthew et al., 2023) solves a text-guided speech-infilling task with large-scale data to learn from context information. However, these methods are trained with single-sentence prompts, lacking an appropriate strategy to extract fine-grained information from multi-sentence speech prompts.

**Prosody Transfer for Speech Synthesis**   Prosody transfer (Lee & Kim, 2019; Klimkov et al., 2019; Gururani et al., 2019; Pan & He, 2021; Karlapati et al., 2022) aims to transfer the prosody from a reference utterance to the synthesized target speech, which is essential for producing natural and expressive speech in a controlled manner (Wagner & Watson, 2010). Skerry-Ryan et al. (2018) first integrate a prosody reference encoder into a TTS system based on Tacotron (Wang et al., 2017), which is capable of performing similar-text prosody transfer. Recent works try to transfer prosody in different-text and different-speaker settings (Karlapati et al., 2020; Zaïdi et al., 2021) with the bottleneck of the prosody encoder. Among them, Daft-Exprt (Zaïdi et al., 2021) uses a gradient reversal layer to penalize the prosody encoder if its output contains information about the speaker identity from the reference utterance, which enhances the target speaker fidelity for cross-speaker prosody transfer. However, as pointed out by Sigurgeirsson & King (2023), current solutions do not learn a transferable representation of prosody, but rather an utterance-level representation that is relatively dependent on both the reference speaker and reference text.

## 3 METHOD

This section introduces Mega-TTS 2. To begin with, we provide an intuitive illustration of how Mega-TTS 2 decomposes the timbre and prosody information from speech. Next, we provide detailed explanations of our prompting mechanisms and the two-stage training process of the proposed model.

### 3.1 DECOMPOSITION FOR PROSODY AND TIMBRE

**Problem Formulation**   Denote $H(X)$ as the Shannon entropy of $X$ and Denote $I(Y; X)$ as the mutual information. We assume that the mel-spectrogram $y$ can be reconstructed through the

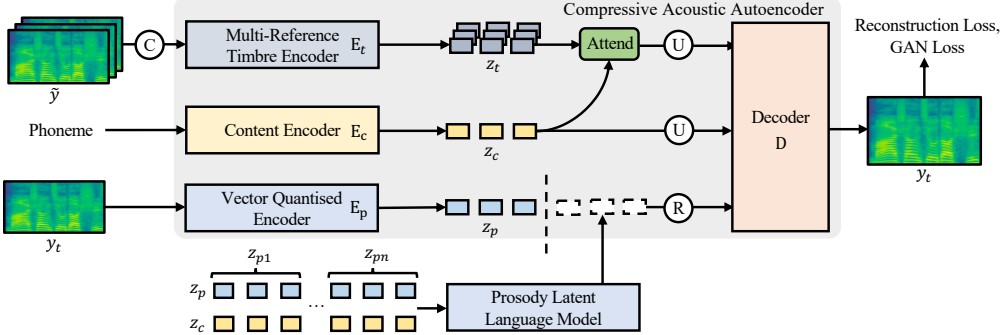

Figure 1: The overall architecture of Mega-TTS 2. ⓒ, ⓤ, ⓡ denotes the concatenation, upsampling, and repeating operations, respectively. $\tilde{y}$ is concatenated along the time axis. "Attend" means the attention operation.

following generative process: $y = D(z_c, z_{pd}, z_t, g)$, where $z_c$ and $z_t$ denote the fine-grained content and timbre hidden states. $g$ denotes the global style information that contains timbre and prosody. We assume that $z_{pd} = (z_p, z_d)$ contains the fine-grained prosodic style information of pitch and energy $z_p$ and duration $z_d$. $z_d = Aligner(y)$ can be obtained by the external alignment tools (McAuliffe et al., 2017) and disentangled from $z_{pd}$. Denote $D$ as the mel-spectrogram decoder. Our goal is to construct an autoencoder-based model to disentangle speech components.

**Decomposition via Corpus Partition** Denote $Y = \{y_1, \cdots, y_n\}$ as the speech corpus for a certain speaker $S$. In training, we partition $Y$ into the target mel-spectrogram $y_t$ and the other mel-spectrograms $\tilde{y}$. Here, we make an important assumption that *the mutual information between $y_t$ and $\tilde{y}$ only contains timbre information $H(z_t)$ and global style information $H(g)$* of $y_t$, i.e.,

$$I(y_t; \tilde{y}) = H(z_t) + H(g) . \tag{1}$$

First, based on the assumption, $z_t$ and $g$ can be extracted through $E_t(\tilde{y})$, and there is no way for $E_t(\tilde{y})$ to obtain the $z_p$ and $z_c$. Second, if we only feed phoneme sequence to $E_c$, $E_c$ can only pass all the content information $z_c$. Third, since the information $z_c$ and $z_t$ are available now, the prosody encoder $E_p$ will prioritize removing the fine-grained content and timbre information if it is forced to lose some information by information bottleneck $B(\cdot)$ (Qian et al., 2019). The bottleneck forces $E_p(y_t)$ to pass only the fine-grained prosodic style $z_p$ that other encoders cannot supply, hence achieving the decomposition. We provide a detailed explanation of how we ensure the validity of Equation 1 in Appendix A.8. After the decomposition, we describe the detailed designs of our prompting mechanisms in the following subsections.

## 3.2 Compressive Acoustic Autoencoder

Note that to store timbre information for thousands of speakers, we need a large number of codebook entries. However, since the prosody and timbre have been decomposed, the prosodic information $z_p$ can be compressed into a highly compact codebook, and the timbre information $z_t$ can be extracted via a powerful speaker encoder. The decomposition strategy not only allows our model to accommodate extremely long prosody prompts but also enables our model to control the prosodic styles of generated speeches. As shown in Figure 1, we design the vector quantised (VQ) encoder as $E_p$, the multi-reference timbre encoder as $E_t$, and the content encoder as $E_c$. Since $E_p$ mainly captures the prosodic variance information, a GAN-based mel-spectrogram decoder $D$ is adopted to model the high-frequency details in spectrograms, which ensures perceptually high-quality reconstructions. Overall, the first-stage training loss can be formulated as $\mathcal{L} = \mathcal{L}_{\text{rec}} + \mathcal{L}_{\text{VQ}} + \mathcal{L}_{\text{Adv}}$, where $\mathcal{L}_{\text{rec}} = \|y_t - \hat{y}_t\|^2$ is the reconstruction loss, $\mathcal{L}_{\text{VQ}}$ is the VQ codebook loss (Van Den Oord et al., 2017), and $\mathcal{L}_{\text{Adv}}$ is the LSGAN-styled adversarial loss (Mao et al., 2017) whose objective is to minimize the distribution distance between the predicted mel-spectrograms and the ground truth mel-spectrograms. Among the proposed three encoders, the content encoder is composed of several feed-forward Transformer layers following common practice in non-autoregressive TTS systems (Ren et al., 2019). In the following paragraphs, we describe the details of the prosody and timbre encoders, respectively.

**Vector Quantised Encoder**   The vector quantised encoder $E_p$ consists of two convolution stacks and a vector quantization bottleneck. The first convolution stacks compress mel-spectrograms into hidden states by a factor of $r$ in length, and the second stacks capture the correlation of features. After that, the vector quantization layer utilizes these hidden states to obtain prosody codes $\mathbf{u} = \{u_1, u_2, ..., u_n\}$ and hidden states $z_p$. The information bottleneck $B(\cdot)$ of the VQ encoder is composed of the temporal compression and the vector quantization layer. The detailed instructions for ensuring an appropriate information bottleneck $B(\cdot)$ can be found in Appendix F.

**Multi-Reference Timbre Encoder**   Our objective is to extract fine-grained timbre information from multi-sentence speech prompts. Since speakers can change their timbre by using different speaking techniques according to their speaking habits or desired semantic meanings (McAdams, 2013), the timbre encoder needs to extract fine-grained timbre information from multiple prompts that can represent the speakers' habits. Here, we introduce a multi-reference timbre encoder (MRTE) to achieve this objective. First, we concatenate the reference mel-spectrograms $\tilde{y}$ that belong to the target speaker but are different from the target mel-spectrogram. The mel encoder then compresses the concatenated mel-spectrogram into acoustic hidden states $z_t$ by a factor of $d$ in length. Subsequently, to extract semantically relevant timbre information from speech prompts, we introduce a timbre-to-content attention module. This module takes $z_c$ as the query and $z_t$ as both the key and the value. Finally, we upsample the output of the timbre-to-content attention module to match the length of the target mel-spectrogram using the length regulator (Ren et al., 2019).

### 3.3 Prosody Latent Language Model

Unlike previous models that are trained with single-sentence prompts, our prosody latent language model (P-LLM) aims to capture the speaker's prosodic patterns from multi-sentence prompts effectively. During the second-stage training process, we first extract the compressed prosody hidden states $\{z_{p1}, z_{p2}, \cdots, z_{pn}\}$ and the content hidden states $\{z_{c1}, z_{c2}, \cdots, z_{cn}\}$ from multiple speech clips $\{s_1, s_2, \cdots, s_n\}$ of the target speaker using the proposed compressive acoustic autoencoder. We then concatenate them along the time axis to construct $z'_p = Concat(z_{p1}, z_{p2}, \cdots, z_{pn})$ and $z'_c = Concat(z_{c1}, z_{c2}, \cdots, z_{cn})$. In order to match the lengths of $z'_p$ and $z'_c$ in the temporal dimension, we expand $z'_c$ to the frame level with duration information $z_d$ and compress it $r$ times with a max pooling layer. After that, we transform $z'_p$ to prosody code $\mathbf{u}'$ and then feed $\mathbf{u}'$ and $z'_c$ into the P-LLM, which predicts the prosody code in an auto-regressive manner:

$$p\left(\mathbf{u}' \mid z'_c; \theta\right) = \prod_{l=0}^{L} p\left(\mathbf{u}'_l \mid \mathbf{u}'_{<l}, z'_c; \theta\right), \tag{2}$$

where $\theta$ is the parameters of P-LLM and $L$ is the length of the concatenated prosody code $\mathbf{u}'$. In training, we set batch size as 1 to increase the maximum number $m$ of prosody codes in each batch as much as possible. If the total number of speech frames from a single speaker is less than $m \times r$, we will include speech samples from other speakers in this batch and incorporate speaker-level attention masks into P-LLM. We do not specifically define the speech prompt; instead, we train the language model directly using the concatenated speech samples through the teacher-forcing technique with the cross-entropy loss. To avoid the transition area problems caused by directly concatenating the prompts, we assign the start token and end token to each sentence, which guides P-LLM to continue writing the current sentence and extract useful information from previous sentences. This training strategy enables the model to capture the useful prosody-level information contained in the multi-sentence prompts. Therefore, in the inference stage, users can flexibly improve the generation quality by extending the length of prompts by concatenating the reference speech clips. For duration modeling, we propose a phoneme-level auto-regressive duration model. This model enhances the duration modeling by leveraging the powerful in-context learning capabilities of auto-regressive models. The overall architecture of the auto-regressive duration model remains the same as P-LLM, but we use mean squared error (MSE) loss instead.

### 3.4 Prosody Interpolation

Here, we propose a prosody interpolation technique to control or replace the prosodic style of the target speaker in the discrete space while ensuring the quality of timbre reconstruction. We achieve this objective by interpolating the probabilities from multiple P-LLM outputs, which come from

multiple speakers. For example, our target speaker has a relatively sad speaking tone, but we want to generate speeches that sound happier for him while preserving his timbre. The solution is to 1) extract prosody latent $\mathbf{u}_a$ from speeches in a happy tone of other speakers and the sad prosody latent $\mathbf{u}_b$ from the target speech prompt; 2) utilize two language models to separately decode the target prosody code $\hat{\mathbf{u}}$ with the prosodic prompt $\mathbf{u}_a$ and $\mathbf{u}_b$. These language models share the same parameters. In every step $t$ of the decoding process, the probability distributions of the two language models are interpolated with the weight $\gamma$, which can be formulated as follows:

$$p\left(\hat{\mathbf{u}}\right) = \prod_{t=0}^{T} \Big( (1-\gamma) \cdot p\left(\hat{\mathbf{u}}_t \mid \hat{\mathbf{u}}_{<t}, \mathbf{u}_b, Concat(z_{cb}, \hat{z}_c); \theta\right) + \gamma \cdot p\left(\hat{\mathbf{u}}_t \mid \hat{\mathbf{u}}_{<t}, \mathbf{u}_a, Concat(z_{ca}, \hat{z}_c); \theta\right) \Big), \quad (3)$$

where $z_{cb}$ and $z_{ca}$ are the content information from speech clips $s_b$ and $s_a$. $\hat{z}_c$ is the content information of the target sentence. With our prosody interpolation technique, users can freely control the prosodic style of the generated speech in the inference stage. Moreover, the proposed prosody interpolation algorithm utilizes the autoregressive probability distribution of the language model for prosody transfer. Compared with directly substituting the time-averaged prosody representation $\mathbf{u}_b$ with $\mathbf{u}_a$ (Karlapati et al., 2020; Zaïdi et al., 2021), the prosody latent language model is able to mix $\mathbf{u}_a$ and $\mathbf{u}_b$ in a soft and fine-grained manner in the autoregressive generation process.

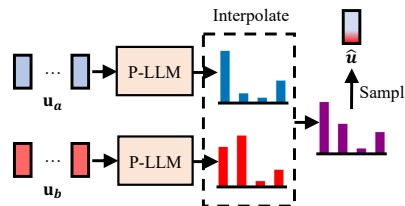

Figure 2: Prosody interpolation.

## 4 EXPERIMENTS

### 4.1 EXPERIMENTAL SETUP

**Training Datasets.** We train Mega-TTS 2 and all baselines on LibriLight (Kahn et al., 2020), which contains 60K hours of unlabelled speech derived from LibriVox audiobooks. The sample rate is 16KHz for all speech data. We transcribe the speech with the hybrid DNN-HMM ASR model pre-trained on 960 hours labeled LibriSpeech following VALL-E (Wang et al., 2023). We align the phoneme sequence with speech using the external alignment tool (McAuliffe et al., 2017).

**Model Configuration.** We provide model configuration in Appendix A.4 and detailed hyper-parameter settings in Table 5.

**Training and Inference.** In the first training stage, we train the first-stage model on 4 NVIDIA A100 GPUs, with a batch size of 48 sentences on each GPU. In the second stage, we train the P-LLM and duration model on 8 NVIDIA A100 GPUs, with a batch size of 4,000 tokens on each GPU. It means that our model supports $4,000 \times 8$ frames of prompts theoretically. We use the Adam optimizer with $\beta_1 = 0.9$, $\beta_2 = 0.999$, $\epsilon = 10^{-9}$ and follow the same learning rate schedule in Vaswani et al. (2017). It takes 600k steps for the first stage model's training and 300K steps for the second stage model's training until convergence. The predicted mel-spectrograms are transformed into audio samples using pre-trained HiFi-GAN V1 (Kong et al., 2020).

**Objective Metrics.** For zero-shot TTS, we evaluate the word error rate (WER), speaker similarity (SIM), and average dynamic time warping (DTW) (Müller, 2007) distance of the pitch for the ground-truth speech and synthesized speech. In terms of the cosine speaker similarity, we use the WavLM model (Chen et al., 2022) fine-tuned for speaker verification[2] to compute the cosine speaker similarity score between the ground-truth speech and the synthesized speech. The similarity score is in the range of $[-1, 1]$, where a larger value indicates a higher similarity of input samples. We also evaluate the word error rate (WER) for cross-lingual TTS. We use the released HuBERT-Large model (Hsu et al., 2021) fine-tuned on the LibriSpeech 960h dataset to transcribe the generated speech into text. Then, the WER between the transcribed text and the original target text is measured. We use all samples in

---

[2]`https://huggingface.co/microsoft/wavlm-base-plus-sv`

Table 1: Zero-shot TTS results on LibriSpeech test-clean set. "-" means the result is not available. "-10s" means that there are 10 seconds of data from each speaker available for fine-tuning or prompting. #Params. denotes the number of parameters (including the vocoder or codec model). The evaluation is conducted with 1 NVIDIA V100 GPU and batch size 1. RTF denotes the real-time factor.

| Model | WER↓ | SIM↑ | DTW↓ | QMOS↑ | SMOS↑ | RTF | #Params. | Method |
|-------|------|------|------|-------|-------|-----|----------|--------|
| *GT* | 1.98% | - | - | $4.43 \pm 0.09$ | $4.26 \pm 0.11$ | - | - | - |
| *Baseline-10s* | 4.74% | 0.895 | 35.12 | $3.97 \pm 0.08$ | $3.76 \pm 0.13$ | | | |
| *Baseline-60s* | 4.18% | 0.923 | 31.08 | $4.01 \pm 0.09$ | $3.92 \pm 0.10$ | 0.089 | 467M | Fine-tune |
| *Baseline-300s* | **3.11%** | **0.934** | **29.80** | **4.08** $\pm$ **0.07** | **4.03** $\pm$ **0.08** | | | |
| *VALL-E-3s* | 5.83% | 0.885 | 36.59 | $3.89 \pm 0.10$ | $3.70 \pm 0.11$ | 1.471 | | |
| *VALL-E-10s* | 5.54% | 0.893 | 34.10 | $3.92 \pm 0.11$ | $3.74 \pm 0.10$ | 1.775 | 478M | Zero-shot |
| *VALL-E-20s* | 8.77% | 0.805 | 43.02 | $3.41 \pm 0.12$ | $3.25 \pm 0.14$ | 2.104 | | |
| *Ours-3s* | 2.46% | 0.898 | 34.39 | $3.99 \pm 0.06$ | $3.75 \pm 0.10$ | 0.302 | | |
| *Ours-10s* | 2.28% | 0.905 | 32.30 | $4.05 \pm 0.08$ | $3.79 \pm 0.09$ | 0.334 | 473M | Zero-shot |
| *Ours-60s* | 2.24% | 0.926 | 30.55 | $4.11 \pm 0.09$ | $3.95 \pm 0.09$ | 0.413 | | |
| *Ours-300s* | **2.23%** | **0.932** | **29.95** | **4.12** $\pm$ **0.10** | **4.01** $\pm$ **0.09** | 0.923 | | |

the test set for the objective evaluation. For prosody transfer, we evaluate the WER, SIM, duration error (DE), and the moments (standard deviation ($\sigma$), skewness ($\gamma$) and kurtosis ($\kappa$)) (Andreeva et al., 2014; Niebuhr & Skarnitzl, 2019) of the pitch distribution.

**Subjective Metrics.** We conduct the MOS (mean opinion score) evaluation on the test set to measure the audio naturalness via Amazon Mechanical Turk. We keep the text content and prompt speech consistent among different models to exclude other interference factors. We randomly choose 50 samples from the test set of each dataset for the subjective evaluation, and each audio is listened to by at least 20 testers. We analyze the MOS in two aspects: QMOS (Quality, clarity, naturalness, and high-frequency details) and SMOS (Speaker similarity in terms of timbre reconstruction and prosodic pattern). We also analyze the CMOS in terms of audio quality and speaker similarity. We tell the testers to focus on one corresponding aspect and ignore the other aspect when scoring.

## 4.2 RESULTS OF ZERO-SHOT SPEECH SYNTHESIS

In this subsection, we evaluate our model with various lengths of speech prompts and compare our model with zero-shot and fine-tuning baselines to demonstrate the effectiveness of the multi-sentence prompting mechanism. We randomly choose 20 speakers from the LibriSpeech test-clean set and randomly choose 400 seconds of speeches for each of them. We split the 400 seconds of speech into a 300-second prompt set and a 100-second target set. We keep the prompts consistent among different models to exclude other interference factors. We compare the zero-shot speech synthesis performance of Mega-TTS 2 with two systems, including: 1) VALL-E (zero-shot) (Wang et al., 2023), a large-scale zero-shot TTS model using large language models to generate discrete speech codes. Since VALL-E has not been open-sourced yet, we carefully implement it for optimal performance; 2) Baseline (fine-tune), a model that incorporates the GAN used in our Mega-TTS 2 to the FastSpeech 2 backbone (Ren et al., 2020). To make the baseline support adaptive scenarios, we use the powerful speaker encoder from Meta-StyleSpeech (Min et al., 2021) to extract timbre information. We carefully fine-tune the baseline system for 2,000 steps to reach an optimal balance between WER and SIM. Note that all of the systems in this experiment are pre-trained on the LibriLight dataset. We provide further explanation for the selection of the baseline systems in Appendix A.7.

**Analysis** As shown in Table 1, as the amount of adaptation data increases, the performance of Mega-TTS 2 continues to improve. Although the performance of VALL-E improves as the data volume increases from 3 seconds to 10 seconds, the performance significantly drops in the 20-second setting due to the single-sentence prompting mechanisms in training. Moreover, since the compression rate of the Encodec model restricts the length of prompts, VALL-E fails to generate reasonable speeches with prompts longer than 20 seconds in our experiments. From another perspective, when we have 10 seconds or 60 seconds of speeches for each speaker, our Mega-TTS 2 surpasses the fine-tuning baseline in terms of speech naturalness and speaker similarity. Additionally, when we have 300

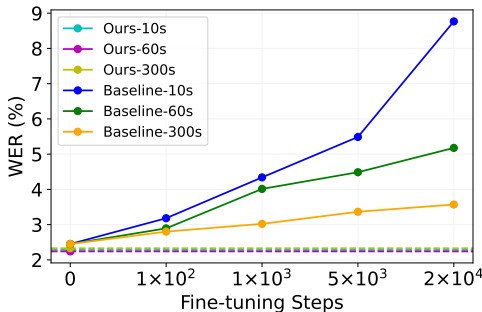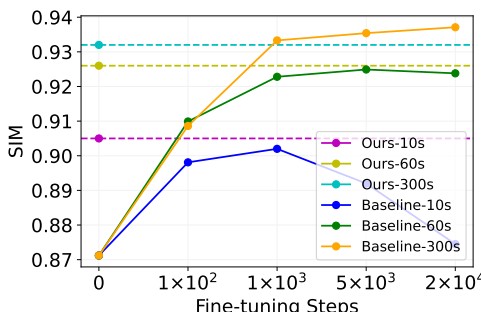

Figure 3: Word error rate (WER) and speaker similarity score (SIM) comparisons in the few-shot fine-tuning process.

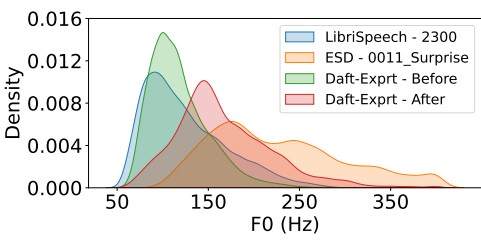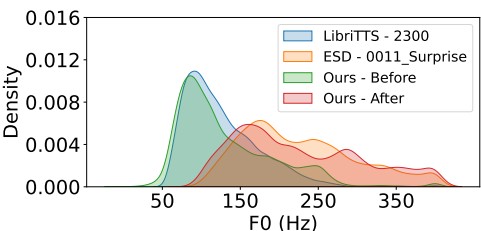

Figure 4: The visualizations of pitch distribution before and after the prosody transfer. Here, we transfer the prosodic styles in a surprised tone to speaker "2300" in the LibriSpeech dataset.

seconds of speeches per speaker, Mega-TTS 2 still outperforms the baseline system in terms of WER and achieves comparable performance with it in terms of speaker similarity. We also visualize the WER and SIM in the fine-tuning process and compare the baseline system with Mega-TTS 2 in Figure 3. Our approach can enhance speaker similarity by utilizing more data like fine-tuning baseline, while maintaining a relatively low word error rate.

## 4.3 RESULTS OF PROSODY TRANSFER

In this subsection, we evaluate the prosody transfer performance of our model by transferring the emotional styles from the ESD dataset (Zhou et al., 2021) to speakers in the LibriSpeech test-clean dataset. We randomly choose 20 speakers from the LibriSpeech test-clean set and choose 50 sentences for each of them. Then, we randomly select an emotional speech clip from the ESD dataset for each of the sentences in the LibriSpeech test-clean set and use the selected emotional speech as the prosodic reference. We keep the reference speeches consistent among different models to exclude other interference factors. We compare the prosody transfer performance of Mega-TTS 2 with two systems, including: 1) CopyCat (Karlapati et al., 2020), a model that utilizes a reference encoder architecture capable of capturing temporal prosodic representations; 2) Daft-Exprt (Zaïdi et al., 2021), a model disentangles identity and prosodic information through an adversarial training strategy that enables accurate prosody transfer across speakers. To make fair comparisons, we incorporate the techniques for prosody transfer from CopyCat and Daft-Exprt to the baseline system proposed in the previous subsection and scale up the model capacity to ensure that all models have a comparable number of parameters. All of the systems in this experiment are pre-trained on the LibriLight dataset.

**Analysis**  Table 2 demonstrates that compared with CopyCat and Daft-Exprt, the moments ($\sigma$, $\gamma$, and $\kappa$) of the generated speeches of Megs-TTS are closer to the ground-truth audio and the DE is lower than other methods, demonstrating the effectiveness of the proposed prosody interpolation techniques. Besides, we observe that our method can efficiently preserve the original timbre and maintain a high audio quality. We also visualize the prosody distribution before and after the prosody transfer process and compare the baseline system with Mega-TTS 2 in Figure 4.

Table 2: Prosody transfer experiments on LibriSpeech test-clean (A) and ESD (B) datasets. SIM-AB means the SIM after we transfer the prosodic styles from B to A. DE denotes the averaged absolute duration error in microseconds. $\sigma$, $\gamma$, and $\kappa$ denotes the moments of the pitch distribution.

| Model | WER | SIM-AB | DE | $\sigma$ | $\gamma$ | $\kappa$ | QMOS | SMOS-AB |
|---|---|---|---|---|---|---|---|---|
| *GT (ESD)* | 4.38% | - | - | 74.91 | 0.707 | 0.024 | $4.18 \pm 0.08$ | 4.22/2.39 |
| *CopyCat* | 5.29% | 0.843/0.740 | 37.2 | 59.74 | 0.889 | 0.859 | $3.72 \pm 0.11$ | 3.53/3.19 |
| *Daft-Exprt* | 4.89% | 0.901/0.633 | 36.5 | 67.20 | 0.851 | 0.427 | $3.90 \pm 0.07$ | 3.81/2.90 |
| *Ours* | **4.82%** | **0.920/0.513** | **32.8** | **72.62** | **0.664** | **0.197** | $3.92 \pm 0.08$ | 3.87/2.64 |

## 4.4 ABLATION STUDIES

**Prosody and Timbre Prompts** We evaluate different lengths of prompts for the MRTE and P-LLM separately. In Table 3, the SIM score and the speech quality increase with longer timbre prompts while the DTW distance almost remains unchanged. When we increase the length of prosody prompts, the DTW distance decreases while the speaker similarity remains at the same level. It can be seen that the proposed timbre and prosody prompting mechanisms boost the subjective speaker similarity in terms of timbre and prosody modeling separately.

Table 3: Ablation studies of the timbre and prosody prompts on zero-shot TTS. "w/ 60s T." denotes that we use 60 seconds of timbre prompts, and "w/ 60s P." denotes that we use 60 seconds of prosody prompts.

| Setting | WER↓ | SIM↑ | DTW↓ | CMOS-Q | CMOS-S |
|---|---|---|---|---|---|
| *Ours-10s* | 2.28% | 0.905 | 32.30 | 0.000 | 0.000 |
| *w/ 60s T.* | 2.26% | 0.922 | 32.23 | +0.128 | +0.241 |
| *w/ 300s T.* | 2.25% | **0.930** | 32.08 | +0.162 | +0.353 |
| *w/ 60s P.* | 2.27% | 0.906 | 30.74 | +0.014 | +0.154 |
| *w/ 300s P.* | 2.24% | 0.908 | **30.25** | +0.017 | +0.196 |

**VQ Encoder and MRTE** We test the following four settings: 1) *w/o MRTE*, which removes the MRTE from our model and does not disentangle the prosody and timbre; 2) *w/ VAE*, which uses VAE to perform generative prosody modeling; 3) *w/ VAE+LDM*, which uses VAE and latent diffusion model (LDM) (Rombach et al., 2022) to perform generative prosody modeling. The architecture and prompting mechanism of LDM is based on Natu-

Table 4: Ablation studies of the proposed VQ encoder and MRTE on zero-shot TTS.

| Setting | WER↓ | SIM↑ | DTW↓ | CMOS-Q | CMOS-S |
|---|---|---|---|---|---|
| *Ours-10s* | 2.28% | 0.905 | 32.30 | 0.000 | 0.000 |
| *Ours-300s* | 2.23% | 0.932 | 29.95 | +0.144 | +0.493 |
| *w/o MRTE* | 5.57% | 0.841 | 36.07 | -0.458 | -0.619 |
| *w/ VAE* | 2.31% | 0.896 | 35.18 | -0.038 | -0.127 |
| *w/ VAE+LDM* | 2.25% | 0.907 | 32.98 | +0.007 | -0.005 |

ralSpeech 2 (Shen et al., 2023b). All baselines use 10 seconds of prompts. The results are shown in Table 4. For setting 1), it can be observed that the removal of MRTE significantly affects both the audio quality and speaker similarity. This is because the timbre information is absorbed by the VQ codebook and puts great pressure on the P-LLM, which demonstrates the effectiveness of decomposing timbre and prosody information. For setting 3), substituting the VQ encoder and P-LLM with VAE and LDM results in similar performance compared to *Ours-10s*. However, the performance of *w/ VAE+LM* is still much inferior to *Ours-300s*, indicating the superiority of the proposed multi-sentence prompting mechanism.

## 5 CONCLUSIONS

In this paper, we present Mega-TTS 2, a framework that boosts the prompting mechanisms for zero-shot TTS systems. With the proposed multi-sentence prompting strategy, our approach outperforms the fine-tuning baseline when 10 seconds to 5 minutes of adaptation data is available for each speaker. Furthermore, our method utilizes a prosody interpolation technique to successfully transfer various prosodic styles to the target speaker while preserving the target speaker's timbre. Experimental results demonstrate that our method exhibits superior performance in terms of audio naturalness and speaker similarity. Due to space limitations, we include additional discussions in the appendix.

## 6 Acknowledgments

This work was supported in part by the National Natural Science Foundation of China under Grant No. 62222211 and National Key R&D Program of China under Grant No.2022ZD0162000.

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

## A  DETAILED EXPERIMENTAL SETTINGS

### A.1  DETAILS IN OBJECTIVE EVALUATIONS

Here, we provide details of the model used in objective evaluations.

**Speaker Similarity Model**   To measure the speaker similarity, we use the WavLM (Chen et al., 2022) model fine-tuned for speaker verification from `https://huggingface.co/microsoft/wavlm-base-plus-sv` to extract the speaker embedding. Then the cosine similarity between the synthesized speech's speaker embedding and the ground-truth speech's speaker embedding is calculated as the speaker similarity score. The WavLM model is pre-trained on 94,000 hours of speech data and fine-tuned on the VoxCeleb1 dataset using an X-Vector head with an Additive Margin Softmax loss, which achieves 0.84%, 0.928%, and 1.758% EER (Equal Error Rate) on the Vox1-O, Vox1-E, and Vox1-H trial lists.

**ASR Model**   To measure the audio quality and speech intelligibility, we evaluate the word error rate (WER) metric. We use the fine-tuned HuBERT-Large model to transcribe the synthesized speech into text and calculate the WER between the transcribed text and the original target text. The HuBERT-Large model from `https://huggingface.co/facebook/hubert-large-ls960-ft` is fine-tuned on 960h of Librispeech and achieves 1.5%, 3.0%, 1.9%, and 3.3% WER on the dev-clean, dev-other, test-clean, and test-other set of Librispeech.

## A.2  DETAILS IN SUBJECTIVE EVALUATIONS

We perform the audio quality and speaker similarity evaluations on Amazon Mechanical Turk (MTurk). For each dataset, we randomly select 50 samples from the test set and use the TTS systems to generate the audio samples. Each audio has been listened to by at least 20 listeners. For MOS, each tester is asked to evaluate the subjective score of a sentence on a 1-5 Likert scale. For CMOS, listeners are asked to compare pairs of audio generated by systems A and B following Loizou (2011), indicating which of the two audio they prefer. For audio quality evaluation (QMOS and CMOS-Q), we tell listeners to "*Please focus on the speech quality in terms of clarity, naturalness, and high-frequency details, and ignore other factors*". For speaker similarity evaluations (MOS-S), we tell listeners to "*Please focus only on the similarity of the speaker to the reference one in terms of the timbre and prosodic patterns, and ignore the differences of content, grammar, audio quality, or other factors.*". We paid $15 to participants hourly and totally spent about $1200 on participant compensation. We tell the participants that the data will be used in scientific research.

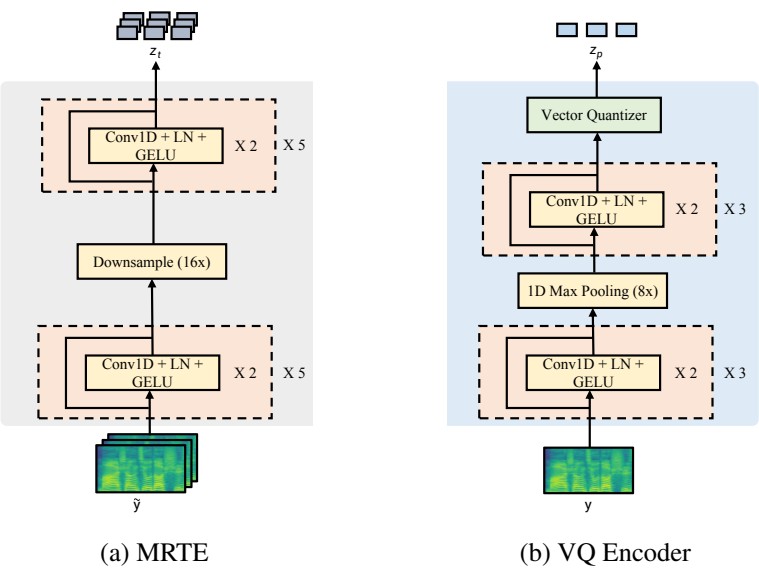

(a) MRTE            (b) VQ Encoder

Figure 5: The multi-reference timbre encoder (MRTE) and vector quantised (VQ) encoder.

## A.3  DETAILED NETWORK STRUCTURE

**MRTE**   As shown in Figure 5, the proposed MRTE is composed of two convolution stacks and a downsampling block. To reduce the computational requirements while maintaining the quality of timbre reconstruction, we downsample the timbre hidden states by a factor of $d = 16$ in length. In training, we randomly sample 2,000 frames from $\tilde{y}$ for training efficiency.

**VQ Encoder**   The bottleneck of our VQ Encoder is composed of a max pooling layer with a stride of 8 and a vector quantised layer. In our experiments, we found that compressing the mel-spectrograms with a compression rate of $r = 8$ yields superior results compared to phoneme-level compression. We have tried different compression rates (2, 4, 8, 16, 32) and found that $r = 8$ reached an optimal balance between the reconstruction performance and compression. On the other hand, in the training process, we also found that the vanilla VQ-VAE suffers from codebook collapse (Takida et al., 2022), which means only a small portion of codebook vectors are optimized. It restricts the expressive capacity of the codebook and affects the convergence of the training process. To solve the codebook collapse issue, we adopt a dynamical initialization strategy based on CVQ-VAE (Zheng & Vedaldi, 2023) during training, which ensures the code vectors that are less-used or unused to be modified more than frequently used ones.

## A.4 MODEL CONFIGURATION

Our Mega-TTS 2 consists of three encoders, a prosody latent language model, a mel decoder, and a discriminator. The prosody encoder, timbre encoder, and decoder consist of 5 convolutional blocks with 512 hidden size and 5 kernel size. The content encoder is an 8-layer Transformer (Vaswani et al., 2017; Shen et al., 2023a) with 512 hidden size. The GAN discriminator follows the architecture of ML-GAN proposed in Chen et al. (2020). The P-LLM model is a decoder-only architecture that contains 12 Transformer layers with 1024 hidden size, which has 151M parameters. The duration predictor is an 8-layer decoder-only Transformer model with 512 hidden size. The codebook embedding size is 1024, and the hidden size of the codebook vector is 256. The compression rate $r$ and $d$ is set as 8 and 16, respectively. For prosody transfer experiments, $\gamma$ is set as 0.8. We provide detailed hyper-parameter settings about the model configuration in Table 5.

Table 5: Hyperparameters of Mega-TTS 2 models.

| Hyper-parameter | | Value |
|---|---|---|
| VQ Prosody Encoder | Encoder Layers | 3 * 2 |
| | Hidden Size | 384 |
| | Conv1D Kernel | 5 |
| | VQ Embedding Size | 1024 |
| | VQ Embedding Channel | 256 |
| Content Encoder | Phoneme Embedding Size | 512 |
| | Encoder Layers | 8 |
| | Hidden Size | 512 |
| | Kernel Size | 5 |
| | Filter Size | 1024 |
| MRTE | Encoder Layers | 5 * 2 |
| | Query Encoder Hidden Size | 512 |
| | Key Encoder Hidden Size | 256 |
| | Key Encoder Stride | 16 |
| | Conv1D Kernel | 3 |
| Mel Decoder | Decoder Layers | 4 |
| | Hidden Size | 512 |
| | Conv1D Kernel | 5 |
| P-LLM | Decoder Layers | 12 |
| | Hidden Size | 1024 |
| | Prosody Code Embedding Size | 1026 |
| | Attention Headss | 16 |
| Multi-Length Discriminator | Number of Discriminators | 3 |
| | Window Size | 32, 64, 128 |
| | Conv2D Layers | 3 |
| | Hidden Size | 192 |
| Total Number of Parameters | | 367M |

## A.5 ERROR BARS AND RANDOM SEEDS

For the subjective evaluations, we report confidence intervals of the results of MOS tests. For the objective evaluations, we ran the experiments 10 times with 10 different random seeds ([1234, 1111, 2222, 3333, 4444, 5555, 6666, 7777, 8888, 9999]) and obtained the averaged results.

### A.6 Sampling Strategy for P-LLM

In all of our experiments, we utilize the top-k sampling strategy for P-LLM, where k is set to 10. The sampling-based method, when used with an appropriate k, enhances the output diversity compared to greedy decoding.

### A.7 About the Selection of Baselines

VALL-E (Wang et al., 2023), NaturalSpeech 2 (Shen et al., 2023b; Leng et al., 2023), and Voice-Box (Matthew et al., 2023) are the state-of-the-art zero-shot TTS models. In the experiments of zero-shot TTS, we have tried to carefully reproduce their works but failed to reproduce NaturalSpeech 2 and VoiceBox. Since all of them do not provide the pre-trained models and source code, we only compare Mega-TTS 2 with VALL-E in our experiments.

### A.8 Detailed Decomposition Strategy

The prosody encoder $E_p$ aims to capture fine-grained and local prosodic style $z_p$. For local prosodic style $z_p$, we assume that $psd(\cdot)$ is a perfect local prosody extractor, and we can obtain the following equation: $I(psd(y_t), psd(\tilde{y})) = 0$. The content information $z_c$ is also local and fine-grained like $z_p$. On the other hand, the global prosodic information like the averaged volume and pitch can not be captured by $E_c$, intuitively. And since we have designed an information bottleneck $B(\cdot)$ for $E_p$, the global prosodic information will be prioritized by the timbre encoder $E_t$ and stored in $H(z_t)$. Now that both the local and global prosodic information is appropriately extracted, the validity of Equation 1 and our disentanglement strategy can be ensured.

## B About Scaling Up Dataset Size

Scaling up dataset size is crucial for the practical application of zero-shot TTS. Therefore, we crawled 200K hours of audiobook recordings from YouTube and novelfm[3]. The crawled corpus contains both labelled and unlabelled speeches, and most of them do not have speaker information. To transcribe the unlabelled speech in the wild, we use a powerful ASR model called WhisperX (Bain et al., 2023). And to obtain the speaker information, we use a released automatic speaker diarization model called *pyannote.audio*[4], which achieves DER=11.24% on the VoxConverse dataset and DER=14.09% on the AISHELL-4 dataset. In this experiment, we do not change the hyperparameter settings of our model. The results are shown in Table 6. It can be seen that increasing the dataset size can improve the speaker similarity of the generated speeches.

Table 6: The results of zero-shot TTS when we scale up dataset size. *Ours-10s (60K) means that we use 60K hours of speeches from LibriLight to train our model and use 10 seconds of speech prompts.*

| Setting | WER↓ | SIM↑ | DTW↓ | CMOS-Q | CMOS-S |
|---|---|---|---|---|---|
| *Ours-10s (60K)* | 2.28% | 0.905 | 32.30 | 0.000 | 0.000 |
| *Ours-10s (200K)* | 2.15% | 0.922 | 32.05 | +0.010 | +0.215 |

## C About the Definition of Adaptive TTS

The concept of adaptive TTS encompasses many aspects like the adaption for different voices, languages, styles, and domains (Tan et al., 2021). It is also known as various terms in academia and industry, such as voice adaptation (Chen et al., 2018), voice cloning (Arik et al., 2018), custom voice (Chen et al., 2021), etc. In this paper, we primarily focus on adaptive TTS for different voices.

---

[3] https://novelfm.changdunovel.com/
[4] https://huggingface.co/pyannote/speaker-diarization

## D    Visualization of Attention Matrices

To further verify the proposed P-LLM and multi-sentence prompting mechanism, we visualize the attention matrices averaged across all layers of P-LLM in Figure 6. In this experiment, we separately conduct short-sentence generation and long-sentence generation. For short-sentence generation, we randomly selected two sentences that are shorter than 3 seconds from speaker "908" in the LibriSpeech test-clean set and concatenated them together. The target texts for Figure 6 (a) and (b) are both about 15 words in length. For long-sentence generation, we randomly selected two sentences that are longer than 15 seconds from speaker "908" in the LibriSpeech test-clean set and concatenated them together. The target texts for Figure 6 (c) and (d) are both about 100 words in length. It can be seen that our P-LLM can capture both short-term and long-term information, demonstrating the effectiveness of the P-LLM's training strategy and the multi-sentence prompting mechanism.

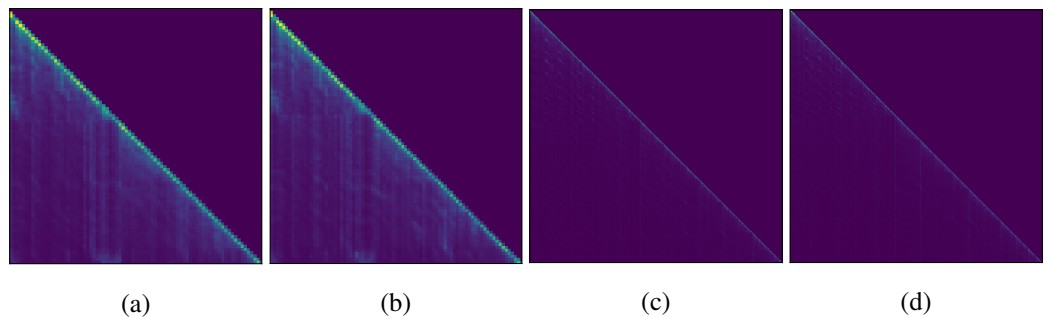

|     (a)     |     (b)     |     (c)     |     (d)     |

Figure 6: Visualization of Attention Matrices. (a) and (b) are the attention matrices in short-sentence generation; (c) and (d) are the attention matrices in long-sentence generation.

## E    Limitations and Ethics Impacts

In this section, we begin by discussing the limitations of the proposed method and outlining our strategies for addressing them in future research. Subsequently, we discuss the ethical impacts that might be brought by zero-shot TTS and our measures to address these concerns.

### E.1    Limitations and Future Work

Firstly, our model is trained on an English dataset and does not support multilingual TTS. We plan to address this problem by introducing more multilingual training data. Secondly, the speech quality can be improved by introducing more high-fidelity training data. Thirdly, a well-designed attention window may further enhance the in-context-learning capability of our P-LLM.

### E.2    Ethics Impacts

Mega-TTS 2 improves the quality and efficiency of zero-shot speech synthesis, which makes it easier for people to synthesize personalized speeches. Under appropriate and legal usage, this technique could facilitate applications like movies, games, podcasts, and other services, making human life more convenient. However, zero-shot TTS may be misused in deepfake-related usages, such as spoofing voices. To handle this, potential solutions like building a corresponding deepfake detection model should be considered. We also plan to add watermarks to the synthesized speeches so that the public can easily tell whether the speeches are synthesized or not. Additionally, restrictions will be included in the license of our project to prevent the misuse of the model.

## F    Description of Information Bottleneck

The settings of the information bottleneck $B(\cdot)$ are crucial for the performance of disentanglement of our proposed method. Intuitively, there are four crucial variables for ensuring an appropriate information bottleneck: the number of codebook embedding, the codebook channel size, the compression

rate $r$ of the VQ Encoder, and the downsampling rate $d$ of the MRTE. However, the search space of these hyperparameters is too large. Since the settings of $r$ and $d$ also influence the reconstruction quality, the burden of P-LLM, and the computational requirements, we first consider $r$ and $d$, fix them, and find the best setting for the hyperparameters of the codebook.

**The Compression Rate** $r$    We have conducted evaluations for different compression rates $r$ of the VQ Encoder. In the experiments, we found that a lower compression rate would result in better reconstruction performance for the compressive acoustic autoencoder, but it would impose a heavier burden on P-LLM since the token sequence is longer. As shown in Table 9, although $r = 2$ achieves the highest objective similarity score, the subjective speech quality and similarity significantly decrease, which means the final quality of generation is affected. Therefore, we use $r = 8$ to reach an optimal balance between the reconstruction performance and compression.

Table 7: Ablation studies of the compression rate $r$. The length of speech prompts is 3 seconds.

| Setting | WER↓ | SIM↑ | DTW↓ | CMOS-Q | CMOS-S |
|---|---|---|---|---|---|
| $r = 2$ | 5.53% | **0.905** | 42.35 | -0.301 | -0.149 |
| $r = 4$ | 4.01% | 0.901 | 36.93 | -0.148 | +0.053 |
| $r = 8$ | **2.46%** | 0.898 | **34.39** | 0.000 | 0.000 |
| $r = 16$ | 2.62% | 0.889 | 35.18 | -0.088 | -0.141 |
| $r = 32$ | 3.57% | 0.880 | 40.98 | -0.288 | -0.252 |

**The Downsampling Rate** $d$    We have conducted evaluations for different downsampling rates $d$ of the MRTE. The results are shown in Figure 10. It can be seen that when the downsampling rate $d$ of the MRTE is low, the mel-spectrogram sequence can provide more information to the timbre encoder, resulting in better reconstruction. However, a low downsampling ratio puts a significant computational burden on the attention operation in MRTE. To reduce the computational requirements while maintaining the quality of timbre reconstruction, we choose $d = 16$ for our Mega-TTS 2.

**The Information Bottleneck with Different Amount of Data**    Intuitively, the performance of the information bottleneck might be very sensitive to the size of the dataset. Therefore, we conduct experiments analyzing the relationship between dataset size and the hyperparameters. The results are presented in Appendix G. Although the hyperparameters do not change across these experiments, we find that the model consistently performs well in scenarios with varying amounts of available data.

## G    SCALING WITH DIFFERENT SIZES OF TRAINING DATA

Here we evaluated the performance of our Mega-TTS 2 scale with varying amounts of available data. In this experiment, all of the systems use 3 seconds of speech prompts. The results are shown in the following table. We can see that Mega-TTS 2 performs well with different sizes of training data, while VALL-E fails to obtain satisfying results when the data is insufficient. We also scale our Mega-TTS 2 with 200K hours of speeches and the results can be found in Appendix B.

## H    THE STRATEGY OF PROSODY MODELING

In this section, we conduct experiments to verify the performance of the phoneme-level, word-level, and stride-8-level prosody modeling. Stride-8 means that the stride of the pooling layer inside the VQ encoder is set to 8. It is worth noting that ProsoSpeech (Ren et al., 2022) utilizes word-level prosody modeling. Both of us use the auto-regressive Transformer for prosody modeling. However, ProsoSPeech aims to improve the naturalness of prosody modeling. Compared with it, our P-LLM aims at improving the similarity of speaker-relevant prosodic patterns, which extracts fine-grained prosodic information from latent prosodic prompts by leveraging the powerful in-context learning capability of LLM. The experimental results are shown in the following table. It can be seen that the stride-8-level prosody modeling achieves the best performance. Intuitively speaking, the phoneme-level prosody modeling provides finer-grained information for better reconstruction while word-level prosody modeling provides more semantic information. Both of these methods would be easily

Table 8: Ablation studies of the compression rate $d$. The length of speech prompts is 3 seconds.

| Setting | WER↓ | SIM↑ | DTW↓ | CMOS-Q | CMOS-S |
|---------|------|------|------|--------|--------|
| $d = 8$ | 2.39% | 0.900 | 34.27 | +0.025 | +0.057 |
| $d = 16$ | 2.46% | 0.898 | 34.39 | 0.000 | 0.000 |
| $d = 32$ | 2.98% | 0.880 | 34.85 | -0.112 | -0.223 |

Table 9: The performance of Mega-TTS 2 scale with different sizes of the training data. All of the systems use 3 seconds of speech prompts.

| Model | Dataset Usage | WER↓ | SIM↑ | DTW↓ | QMOS | SMOS |
|-------|---------------|------|------|------|------|------|
| VALL-E | LibriLight (1k hours) | 15.73% | 0.782 | 54.05 | $3.63_{\pm0.13}$ | $3.52_{\pm0.10}$ |
| | LibriLight (10k hours) | 8.29% | 0.870 | 38.93 | $3.84_{\pm0.12}$ | $3.65_{\pm0.12}$ |
| | LibriLight (60k hours) | 5.83% | 0.885 | 36.59 | $3.90_{\pm0.12}$ | $3.71_{\pm0.10}$ |
| Mega-TTS 2 | LibriLight (1k hours) | 4.69% | 0.861 | 43.07 | $3.88_{\pm0.11}$ | $3.61_{\pm0.10}$ |
| | LibriLight (10k hours) | 2.91% | 0.882 | 35.76 | $3.95_{\pm0.11}$ | $3.70_{\pm0.12}$ |
| | LibriLight (60k hours) | 2.46% | 0.898 | 34.39 | $3.98_{\pm0.09}$ | $3.77_{\pm0.08}$ |

afftected by the alignment accuracy of the speeches in training and inference stages. In order to enhance the stability and performance of the proposed model, we use stride-8-level prosody modeling.

Table 10: Ablation studies of the strategy of prosody modeling. The length of speech prompts is 3 seconds.

| Setting | WER↓ | SIM↑ | DTW↓ | CMOS-Q | CMOS-S |
|---------|------|------|------|--------|--------|
| Stride-8-level | 2.46% | 0.898 | 34.39 | 0.000 | 0.000 |
| Phoneme-level | 2.94% | 0.897 | 35.78 | -0.107 | -0.052 |
| Word-level | 3.12% | 0.892 | 34.42 | -0.120 | -0.078 |

# I  SPECIAL CASES FOR ASSUMPTION 1

In practical scenarios, there is a special case for Assumption 1: the timbre of a speaker may vary significantly over different time periods. To address this special case, we select $\tilde{y}$ randomly from regions near $y_t$ as much as possible, ensuring that the timbre information of $\tilde{y}$ is close to that of $y_t$.

# J  DIFFERENT LENGTHS OF CONTEXT DURING TRAINING

Here we make ablation studies for different lengths of context during our model's training process. We separately train P-LLM with different numbers of the contextual VQ code tokens and train our compressive acoustic autoencoder with different numbers of the contextual mel-spectrogram frames for MRTE. The results are shown in Figure 7. It can be seen that when we increase the length of context, the performance of the model during training significantly improves, demonstrating the effectiveness of our multi-reference training strategy.

# K  EXPLANATIONS ABOUT MORE CASES OF ROBUST SPEECH SYNTHESIS

In our Mega-TTS 2, we employ a language model only for prosody modeling, enabling our model to benefit from the advantages of in-context learning provided by the LLM model. This approach also helps to address the robustness issues (word skipping or repeating) associated with the autoregressive TTS model. Therefore, we make explanations about more cases of robust speech synthesis, to demonstrate our method's necessity. In commercial scenarios, news reporting, and other formal scenarios, robustness is a crucial factor. Just a few repeating or skipping words can have significant negative impacts. These situations are better suited for models with duration models that ensure robustness, such as FastSpeech [4], Glow-TTS [5], and our Mega-TTS 2. However, for models like tacotron [6], word omissions or repetitions can significantly affect the listening experience. On

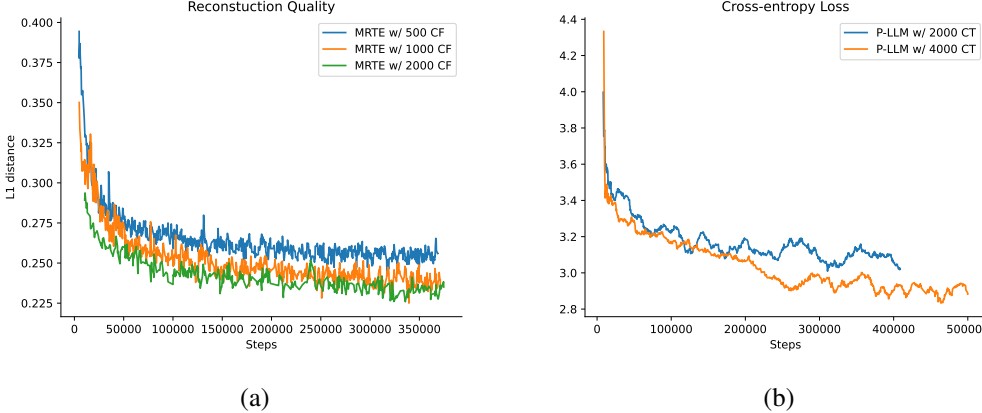

(a)                                          (b)

Figure 7: Visualization of training loss curves with different lengths of context. In subfigure (a), "MRTE w/ 500 CF" means we set the number of the contextual mel-spectrogram frames for MRTE to 500. In subfigure (b), "P-LLM w/ 2000 CT" means we set the number of the contextual VQ code tokens for P-LLM to 2000.

the other hand, in some scenarios, robustness is relatively less important. For example, occasional missing words or repetitions in dialogue scenes can also be natural.

## L  RESULTS WITH NOISY REFERENCE PROMPTS

To verify our model's robustness against noisy reference prompts, we conduct experiments on LibriSpeech test-other set. The experimental setup for this experiment is consistent with the one described in Section 4.2. The results are shown in Table 11. It can be seen that Mega-TTS 2 maintains excellent performance with noisy reference prompts.

Table 11: Results of zero-shot TTS on the LibriSpeech test-other set.

| Model | WER↓ | SIM↑ | DTW↓ | QMOS | SMOS |
|---|---|---|---|---|---|
| *GT* | 3.17% | - | - | $3.95_{\pm0.09}$ | $3.98_{\pm0.08}$ |
| *VALL-E-3s* | 6.21% | 0.889 | 36.13 | $3.64_{\pm0.12}$ | $3.70_{\pm0.11}$ |
| *Ours-3s* | **2.73%** | **0.904** | **32.50** | $\mathbf{3.77_{\pm0.10}}$ | $\mathbf{3.80_{\pm0.09}}$ |

