# OpenReview forum: "Mega-TTS 2: Boosting Prompting Mechanisms for Zero-Shot Speech Synthesis"
_ICLR.cc/2024/Conference — ICLR 2024 poster_

### Official Review · Reviewer_WuFJ · 2023-10-28

**Soundness:** 3 good
**Presentation:** 3 good
**Contribution:** 2 fair
**Rating:** 6
**Confidence:** 5

**Summary:**

This work proposed a zero-shot Text-to-speech model with a prosody large language model (p-LLM). They utilize a multi-reference timbre encoder to extract a timbre from multiple references. Moreover, they introduce an autoregressive duration predictor for prosody modeling. The results show that they can transfer the prosody and timbre respectively.

**Strengths:**

They organize the parallel TTS pipeline with autoregressive representation modeling. They only utilize a large language model in prosody modeling so they could enjoy the advantage of in-context learning of the LLM model and may prevent a problem from the autoregressive TTS model which could synthesize a speech with repeating and skipping. It would be better if the authors could explain more cases of robust speech synthesis without repeating and skipping..

In my opinion, autoregressive duration predictor could significantly improve the prosody transfer performance because duration influences prosody. It would be grate if you could add additional ablation study for autoregressive and non-autoregressive duration predictor.

Meanwhile, an adversarial duration predictor was proposed in VITS2. This could improve the performance of duration modeling in this work.

**Weaknesses:**

1.	I just wonder why the authors do not state ProsoSpeech which has the same structure as P-LLM without word-level prosody modeling. It would be better if the authors add the ablation study for phoneme and word-level prosody modeling.

2.	The authors should have conducted more experiments on prosody modeling. There are recently prosody modeling works, Prosody-TTS and CLAPSpeech. Although recently large language model has been investigated, I hope that the author could add an additional experiment with them. All of the papers including this work might be from the same research group but they did not state anything. I hope the authors address this issue.

3.	Multi-reference Style Transfer methods were already utilized in many speech papers such as Attentron [S. Choi, 2020].

[S. Choi, 2020] S Choi, “Attentron: Few-Shot Text-to-Speech Utilizing Attention-Based Variable-Length Embedding,” Interspeech 2020

**Questions:**

1.	I have a question about the Baseline model. This model might be a FastSpeech 2 with GAN training and style encoder of meta-stylespeech. Why do you fine-tune this model? I just wonder about the performance of zero-shot TTS of baseline.

2.	Are there any failure cases for the auto-regressive duration predictor? In my opinion, there are many components of this work. I think AR duration predictor is one of the important changes and it could be utilized for other TTS model easily so I hope you to analyze this predictor with additional ablation studies. Do not need to train the model additionally. I just suggest to infer the baseline TTS model with the predicted duration of your model.

3.	Does the model synthesize the speech robustly with the noisy reference prompt? I would be better if you could add the results on test-other set.

Although recently methods introduce codec-based speech synthesis, it is also important to utilize conventional acoustic representation such as Mel-spectrogram so I like the concepts of this paper adopting the language model for prosody modeling. However, I hope the authors include additional experiments

1. Results on conditioning noisy reference prompts

2. Comparisons with other prosody modeling methods such as Prosody-TTS and CLAPSpeech

3. Comparison with word-level Prosody modeling (ProsoSpeech)

---

> ### Author Response · Authors · 2023-11-19
> **Response to Reviewer WuFJ (Part 1/3)**
>
> We thank the reviewer for the constructive and professional review. We hope our response resolves your concerns fully.
>
> **[About the missing statement of ProsoSpeech and the missing ablation study for phoneme-level and word-level prosody modeling]**
> We are sorry for the missing statement of ProsoSpeech. Here, we provide detailed descriptions of the differences between ProsoSpeech and our method. We also conduct ablation studies for the prosody modeling strategies. These descriptions and results are also added to Appendix H.
>
> *Differences:* We agree that ProsoSpeech [1] has the same structure as P-LLM proposed in this paper (both of us use the auto-regressive Transformer for prosody modeling). However, ProsoSpeech aims to improve the **naturalness** of prosody modeling for traditional TTS, which is only conditioned on text and global speaker information. Compared with it, our P-LLM aims at improving the **similarity** of speaker-relevant prosodic patterns for zero-shot TTS, which extracts fine-grained prosodic information from latent prosodic prompts by leveraging the powerful in-context learning capability of LLM. We also propose a novel training strategy that enables P-LLM to capture the useful prosody-level information contained in the multi-sentence contextual prompts. Moreover, our work mainly aims at boosting the timbre and prosody prompting mechanisms for zero-shot TTS.
>
> *Ablation study for prosody modeling strategy:* We use stride-8-level prosody modeling (the stride of the pooling layer inside the VQ encoder is set to 8) and ProsoSpeech uses word-level prosody modeling. We agree that the ablation studies in this aspect are very important. Therefore, we conducted the following experiments to verify the performance of the phoneme-level, word-level, and stride-8-level prosody modeling. In this experiment, all of the systems use 3 seconds of prompts. The results are shown in the following table. It can be seen that the stride-8-level prosody modeling (used in our paper) achieves the best performance. Intuitively speaking, the phoneme-level prosody modeling provides finer-grained information for better reconstruction while word-level prosody modeling provides more semantic information. Both of these methods would be easily affected by the alignment accuracy of the speeches in training and inference stages. In order to enhance the stability and performance of the proposed model, we use stride-8-level prosody modeling.
>
> | Setting        |  WER  |  SIM  |  DTW  | CMOS-Q | CMOS-S |
> | -------------- | :---: | :---: | :---: | :----: | :----: |
> | Stride-8-level | 2.46% | 0.898 | 34.39 | 0.000  | 0.000  |
> | Phoneme-level  | 2.94% | 0.897 | 35.78 | -0.107 | -0.052 |
> | Word-level     | 3.12% | 0.892 | 34.42 | -0.120 | -0.078 |
>
>
>
> **[About the missing comparisons with other prosody modeling methods such as Prosody-TTS and CLAPSpeech]**
> Although our Mega-TTS, Prosody-TTS, and CLAPSpeech are all prosody modeling methods, the objectives and tasks of our Mega-TTS are different from theirs:
>
> - Prosody-TTS [2], and CLAPSpeech [3] aim at improving the overall prosody naturalness for traditional TTS, while our work focuses more on improving the prosody similarity for Zero-shot TTS. Prosody-TTS [1] measures the MOS-P (**naturalness** of pitch, energy, and duration) as one of their metrics. While, as illustrated in Section 4.1 and Appendix A.2, we use SMOS to measure the prosody and timbre **similarity** for each speaker.  CLAPSpeech aims to improve the prosody prediction by encouraging the text encoder to extract prosody variance from the text context with contrastive loss in the joint multi-modal space. The text encoder proposed in CLAPSpeech can be seen as speaker-irrelevant. Moreover, the text encoder proposed in CLAPSpeech can be also used in our work as a plugin module.
>
> Since the objectives and tasks of Prosody-TTS and CLAPSpeech are different from the task of our Mega-TTS, we did not make comparisons with Prosody-TTS, and CLAPSpeech in our paper.

---

> > ### Author Response · Authors · 2023-11-19
> > **Response to Reviewer WuFJ (Part 2/3)**
> >
> > **[About the reasons for fine-tuning the baseline and the performance of zero-shot TTS of the baseline]**
> > We fine-tune this model to construct a strong baseline. The performance of zero-shot TTS of the baseline with 10 seconds of prompt is shown in the following table. It can be found that although the zero-shot baseline slightly outperforms the baseline fine-tuned with 10 seconds of data in terms of speech quality, the speaker similarity of the zero-shot baseline is significantly lower than that of the other methods. Therefore, we choose the fine-tuned version of the baseline system as a strong baseline for zero-shot TTS.
> >
> > | Setting                 |  WER  |  SIM  |  DTW  | CMOS-Q | CMOS-S |
> > | ----------------------- | :---: | :---: | :---: | :----: | :----: |
> > | Baseline zero-shot      | 3.26% | 0.871 | 40.41 | 0.000  | 0.000  |
> > | Baseline 10s fine-tune  | 4.74% | 0.895 | 35.12 | -0.182 | +0.358 |
> > | Baseline 60s fine-tune  | 4.18% | 0.923 | 31.08 | -0.101 | +0.462 |
> > | Baseline 300s fine-tune | 3.11% | 0.934 | 29.80 | +0.056 | +0.517 |
> >
> >
> >
> > **[About the proposed auto-regressive duration predictor]**
> > *Question:* Are there any failure cases for the auto-regressive duration predictor?
> >
> > *Answer:* In our experiments, the robustness of the proposed auto-regressive duration predictor is comparable to that of the one used in the baseline TTS model (the duration predictor proposed in FastSpeech 2 [8]), and both models have very few failure cases. As shown in Table 1 in our paper, our Mega-TTS 2 achieves the lowest WER score, demonstrating its robustness.
> >
> >
> >
> > *About additional experiments:* Thanks for your valuable and constructive suggestions about our auto-regressive duration predictor! We conduct additional experiments to demonstrate that our auto-regressive duration predictor can be utilized for other TTS model easily as a plugin module. Here we infer the zero-shot version of the baseline TTS models with the predicted duration of our auto-regressive duration predictor. The results are shown in the following table. It can be seen that both the speech naturalness and the prosody similarity are improved by our auto-regressive duration predictor.
> >
> > | Setting                            | Duration Error | CMOS-Q | CMOS-S |
> > | ---------------------------------- | :------------: | :----: | :----: |
> > | Duration Predictor in FastSpeech 2 |     27.39      | 0.000  | 0.000  |
> > | Auto-regressive Duration Predictor |     25.78      | +0.071 | +0.103 |
> >
> >
> >
> > *About the duration predictor in VITS2 [7]:* Thank you for your suggestions about adopting the adversarial duration predictor proposed in VITS2. Intuitively, both the autoregressive duration predictor in our work and the adversarial duration predictor in VITS2 are powerful generative models suitable for duration modeling (a speech attribute with rich diversity). We will try the adversarial duration predictor in VITS2 in our future work.
> >
> > **[About the multi-reference style transfer methods]**
> > We apologize for not discussing the multi-reference style transfer methods in the previous version of the paper. In the revised edition, we discussed these methods in detail and highlighted the main differences between them and Mega-TTS in Section 2 (marked in red).

---

> ### Author Response · Authors · 2023-11-19
> **Response to Reviewer WuFJ (Part 3/3)**
>
> **[About the results on conditioning noisy reference prompts]**
> Here we conduct zero-shot TTS experiments on LibriSpeech test-other set to verify our model's robustness with noisy reference prompts. From the following table, we can see that Mega-TTS maintains excellent performance with noisy reference prompts. We have added these results to Appendix L in the new version of the paper (marked in red). The generated speech samples can be accessed in the "Additional Examples for Rebuttal" Section of our demo page: [https://boostprompt.github.io/boostprompt/](https://boostprompt.github.io/boostprompt/).
>
> | Model       |    WER    |    SIM    |    DTW    |       QMOS        |       SMOS        |
> | ----------- | :-------: | :-------: | :-------: | :---------------: | :---------------: |
> | GT          |   3.17%   |     -     |     -     |   3.95$\pm$0.09   |   3.98$\pm$0.08   |
> | VALL-E-3s   |   6.21%   |   0.889   |   36.13   |   3.64$\pm$0.12   |   3.70$\pm$0.11   |
> | Mega-TTS-3s | **2.73%** | **0.904** | **32.50** | **3.77$\pm$0.10** | **3.80$\pm$0.09** |
>
>
>
> **[Explanations about more cases of robust speech synthesis]**
> In commercial scenarios, news reporting, and other formal scenarios, robustness is a crucial factor. Just a few repeating or skipping words can have significant negative impacts. These situations are better suited for models with duration models that ensure robustness, such as FastSpeech [4], Glow-TTS [5], and our Mega-TTS. However, for models like tacotron [6], word omissions or repetitions can significantly affect the listening experience. On the other hand, in some scenarios, robustness is relatively less important. For example, occasional missing words or repetitions in dialogue scenes can also be natural. Thanks for your advice. We have added these explanations to Appendix K.
>
> **[References]**
> *[1] Ren, Yi, et al. "Prosospeech: Enhancing prosody with quantized vector pre-training in text-to-speech." ICASSP 2022-2022 IEEE International Conference on Acoustics, Speech and Signal Processing (ICASSP). IEEE, 2022.*
> *[2] Huang, Rongjie, et al. "Prosody-TTS: Improving Prosody with Masked Autoencoder and Conditional Diffusion Model For Expressive Text-to-Speech." Findings of the Association for Computational Linguistics: ACL 2023. 2023.*
> *[3] Ye, Zhenhui, et al. "CLAPSpeech: Learning Prosody from Text Context with Contrastive Language-Audio Pre-training." arXiv preprint arXiv:2305.10763 (2023).*  *[4] Ren, Yi, et al. "Fastspeech: Fast, robust and controllable text to speech." Advances in neural information processing systems 32 (2019).*
> *[5] Kim, Jaehyeon, et al. "Glow-tts: A generative flow for text-to-speech via monotonic alignment search." Advances in Neural Information Processing Systems 33 (2020): 8067-8077.*
> *[6] Wang, Yuxuan, et al. "Tacotron: Towards end-to-end speech synthesis." arXiv preprint arXiv:1703.10135 (2017).*
> *[7] Kong, Jungil, et al. "VITS2: Improving Quality and Efficiency of Single-Stage Text-to-Speech with Adversarial Learning and Architecture Design." arXiv preprint arXiv:2307.16430 (2023).*
> *[8] Ren, Yi, et al. "Fastspeech 2: Fast and high-quality end-to-end text to speech." arXiv preprint arXiv:2006.04558 (2020).*

---

> ### Comment · Reviewer_WuFJ · 2023-11-19
> **Thanks for your response**
>
> The authors addressed my concerns about the issues I raised. Thanks for the detailed explanation. I will raise the score from 5 to 6.
>
> By the way I have a question for prosody modeling. I have noticed the authors utilized a fixed frame information bottleneck like AUTOVC, not phoneme-level modeling so I have a concern about the proper size of information bottleneck. In my experience, this proper size of autovc may differ according to the sampling rate and hop size so this makes it difficult to reproduce the results if the preprocessing methods or dataset are changed. In this paper, they adopt this information bottleneck on the phoneme sequence so it would be better if you could add the results or your experiences for proper information bottleneck size according to different language dataset or type of text sequence (grapheme or phoneme).

---

> ### Author Response · Authors · 2023-11-21
> **Dear Reviewer WuFJ**
>
> We are grateful to receive your feedback. In AutoVC, the fixed frame information bottleneck needs to be carefully adjusted to separate the content information from the source speech, making it difficult to reproduce the results if the preprocessing methods or datasets are changed. However, in this work, we directly provide the text content to our model and we only adopt this information bottleneck for the prosody encoder (not for the content and timbre encoder). The objective for disentanglement is the prosody patterns. As a result, the information bottleneck in our work is comparatively more robust and suitable for various datasets.

---

> ### Comment · Reviewer_WuFJ · 2023-11-22
> **Thanks for your response**
>
> Thanks for your response. I agree that the information bottleneck on prosody representation is more robust. I hope to discuss more at Vienna thank you, and this is my last question, do you have a plan to release the source code of your model?

---

> > ### Author Response · Authors · 2023-11-22
> > **Thanks for your careful review and the helpful discussion**
> >
> > Thanks for your careful review and the helpful discussion! Although there are many exciting use cases for zero-shot TTS models, the misuse of them will lead to many social security issues. After conducting evaluations for our Mega-TTS model, we found that the majority of people are unable to distinguish between real and generated fake speeches. Releasing it would pose potential risks to social security and personal rights. Therefore, the pre-trained weights or code of our Mega-TTS can not be open-sourced at this time. To strike the right balance between openness with responsibility, we are still trying to solve security problems through robust watermarking and other algorithms. After we have resolved the security issues, we genuinely look forward to inviting you to experience our voice cloning open API in Vienna.

---

### Official Review · Reviewer_azu9 · 2023-10-30

**Soundness:** 3 good
**Presentation:** 4 excellent
**Contribution:** 4 excellent
**Rating:** 6
**Confidence:** 4

**Summary:**

This paper proposes a novel zero-shot TTS framework that can effectively disentangle and control prosody and timbre with extremely long prompts. Specifically, a multi-reference timbre encoder is proposed to extract the timbre information, and a P-LLM is proposed to generate prosody with multiple reference context. A prosody transferring technique is proposed to control the generated prosody with context. Extensive experiments are done to show the superior performance of the proposed method.

**Strengths:**

1. The proposed TTS framework can separately control both timbre and prosody. Especially the zero-shot prosody control, which is one of the most challenging topic in TTS area.
2. The proposed method can scale the in-context learning to very long prompts, like 300s, and the performance does not saturate when prompt is longer than 20s, which is promising.
3. Although verifying the superiority of a zero-shot TTS system is hard given that most of them are closed-sourced, the authors reimplemented the baseline methods and do the comparison with controlled variables like parameter number, and training datasets.

**Weaknesses:**

1. Some of the zero-shot TTS categories are missing from the related works. One of the most related is the attention-base adaptation method. Early in year 2020, Attentron [1] is proposed, which can adapt to unseen speakers with multiple reference utterances, just like the MRTE in MegaTTS. Such strategy is also used in zero-shot voice conversion domain [2]. Later, methods like [3] introduces cross-attention based model and compress the reference into a fixed-length sequence before decoder attention, which is more close to the one in NaturalSpeech2.
2. Some ablation studies are missing. Since the primary design of the training strategy is using multiple reference instead of only one utterance, it is important to show the performance difference between using long context vs short context during training to verify the necessity of this training strategy.

[1] Choi, Seungwoo, et al. "Attentron: Few-shot text-to-speech utilizing attention-based variable-length embedding." Interspeech 2020.

[2] Lin, Yist Y., et al. "Fragmentvc: Any-to-any voice conversion by end-to-end extracting and fusing fine-grained voice fragments with attention." ICASSP 2021-2021 IEEE International Conference on Acoustics, Speech and Signal Processing (ICASSP). IEEE, 2021.

[3] Yin, Dacheng, et al. "RetrieverTTS: Modeling decomposed factors for text-based speech insertion." Interspeech 2022

**Questions:**

1. Is there a justification of the assumption in section 3.1 "the mutual information between $y^t$ and $\tilde{y}$ only contains timbre information"? This assumption is not very obvious, since in the audiobook dataset, some performing skills may change the timbre largely in different sentences. Additionally, the average prosody style information can also be shared by different utterances.
2. Is it possible to show the baseline methods with 300s context?

---

> ### Author Response · Authors · 2023-11-19
> **Response to Reviewer azu9**
>
> We are grateful for your positive review and valuable feedback, and we hope our response fully resolves your concern.
>
> **[About the missing zero-shot TTS categories in the related works]**
> We are sorry that the attention-base adaptation method is missing in the related works. In the revised version of the paper, we have included detailed discussions on the attention-based adaptation methods and highlighted the main differences between them and our own approach in Section 2 (marked in red). We would like to express our gratitude for your valuable suggestions once again.
>
> **[About the performance difference between using long context vs short context during training]**
> Yes, the ablation studies of the context length during training are essential. We have separately trained our Mega-TTS with different numbers of the contextual VQ codes for P-LLM and trained our Mega-TTS with different numbers of the contextual mel-spectrogram frames for MRTE. After evaluating the reconstruction quality for our compressive acoustic autoencoder and the cross-entropy loss for our P-LLM, we find that the performance of the model significantly **improves when we increase the length of context**, demonstrating the effectiveness of our multi-reference training strategy. We have added the experimental results and analysis to Appendix J in the new version of the paper (marked in red).
>
>
>
> **[About the assumption in section 3.1] (Question 1)**
> Yes, we agree that: 1) some performing skills may change the timbre largely in different sentences; 2) the the global style information can also be shared by different utterances. For case 1), we select $\tilde{y}$ randomly from regions near $y_{t}$ as much as possible, ensuring that the timbre information of $\tilde{y}$ is close to that of $y_{t}$. We have made descriptions of it in Appendix I in the new version of the paper (marked in red). For case 2), under this circumstance, the timbre encoder may capture **global style information**. However, the **local prosodic patterns** vary dramatically across different utterances by the same speaker. The P-LLM in our paper mainly aims at capturing the **fine-grained (local) prosodic patterns**. To enhance the clarity of our paper, we have provided additional clarification for assumption (1) and explained that we only disentangle the fine-grained prosody and timbre information in the new version of the paper (marked in red). Furthermore, our experiments demonstrate the capability of Mega-TTS to independently prompt prosody and timbre in a zero-shot setting. Even in some emotional datasets (the ESD dataset) and in some in-the-wild corpus (as shown in the demo page), our method is also able to clone non-audiobook style voices effectively.
>
>
>
> **[About the baseline methods with 300s context]**
> There are two types of baseline methods in our work: 1) VALL-E; 2)  a baseline that incorporates the GAN used in our Mega-TTS to the FastSpeech 2
> backbone [1].
>
> 1. For VALL-E, it is impossible to show the results with 300s context. Due to the low compression rate of the Encodec model, the maximum context is limited (around 30 seconds) during training. When the length exceeds this limit during inference, the generation quality will be very poor. As shown in "long-form zero-shot TTS experiments" on our demo page, VALL-E fails to generate intelligible results with long context.
> 2. For the baseline 2), it is fine-tuned on the corresponding context. Therefore, for Baseline-300s, it has actually encountered and learned the speech information from each speaker's 300-second context, which is more powerful than directly feeding the 300s context into the reference encoder and encoding it to a time-averaged global vector.
>
>
>
> **[References]**
> *[1] Ren, Yi, et al. "Fastspeech 2: Fast and high-quality end-to-end text to speech." arXiv preprint arXiv:2006.04558 (2020).*

---

> ### Comment · Area_Chair_cyXG · 2023-11-22
> **Reminder to respond to authors' rebuttal**
>
> Dear Reviewer,
>
> Please respond to authors rebuttal and see whether they have addressed your concerns. Thanks!

---

### Official Review · Reviewer_6vRj · 2023-11-02

**Soundness:** 3 good
**Presentation:** 2 fair
**Contribution:** 3 good
**Rating:** 6
**Confidence:** 3

**Summary:**

This paper introduces Mega-TTS, a zero-shot TTS framework designed to enhance multi-sentence prompts by decomposing them into timbre and prosody information. Mega-TTS utilizes an acoustic auto-encoder to independently encode prosody, content, and timbre information. The model integrates a multi-reference timbre encoder and a prosody latent language model (P-LLM) for efficient extraction of information from multi-sentence prompts. This design facilitates the generation of transferable and controllable prosody by leveraging probabilities derived from P-LLM outputs. The paper demonstrates that the synergy between the multi-reference timbre encoder and the prosody interpolation enabled by P-LLM results in fine-grained and controllable prosody transfer. The proposed outperforms Vall-e and the fine-tuning baseline when speech prompts ranging from 10 seconds to 5 minutes are used.

**Strengths:**

- The proposed method adeptly combines the advantages of non-autoregressive (non-AR) modeling, such as robustness and controllability, with the powerful expressiveness of auto-regressive (AR) modeling, achieving this by decomposing speech into prosody and timbre using an information bottleneck.
- The proposed model exhibits the capability to independently prompt prosody and timbre within a zero-shot setting.
- The proposed method empirically shows improved zero-shot performance compared to fine-tuning approaches and outperforms existing state-of-the-art models, and the prosody transfer by using a prosody interpolation technique.

**Weaknesses:**

- While we understand the issue of having a limited number of available baselines, the absence of comparisons with NaturalSpeech2 and VoiceBox makes it challenging to ascertain the proposed model's superiority. At the very least, it appears necessary to replicate the experimental conditions described in the baseline papers and evaluate the proposed model, comparing its performance using the metrics reported in each paper, especially for prompts that are 3 seconds long. It raises the question: How does the proposed method perform with 3-second prompts?
- In the case of datasets like LibriVox, which comprises audiobooks, it is commonly observed that there is not a significant variation in speaking style across different utterances by the same speaker. In this context, it becomes difficult to uphold assumption (1). As a result, the timbre encoder may capture prosody information as well, and the experiment does not conclusively demonstrate the complete separation of these two elements.
- The description of the information bottleneck, a crucial component of the proposed method, is lacking in detail. Specifically, there is an absence of clear guidelines and processes for setting variables such as $r$, $d$, and the hidden dimensions to ensure an appropriate information bottleneck.
- From the perspective of a practitioner, the prerequisite of using Montreal Forced Aligner (MFA) to extract alignments beforehand could be seen as a cumbersome step.

**Questions:**

- What would be the performance outcome if we generate the prosody latent for the target speaker using only the prosody information from another speaker, instead of using interpolation?
- It is unclear in which dimension the concatenation is performed in the Multi-Reference Timbre Encoder. While Figure 1 seems to suggest that concatenation occurs along the hidden dimension axis, the description in Section 3.2 of the timbre-to-content attention module implies that it happens along the time axis.
- In the comparison experiments with VALL-E, why was a different model chosen to measure speaker similarity (SIM) instead of using WavLM the one utilized [1]?

[1] Wang, Chengyi, et al. "Neural codec language models are zero-shot text to speech synthesizers." arXiv preprint arXiv:2301.02111 (2023).

---

> ### Author Response · Authors · 2023-11-19
> **Response to Reviewer 6vRj (Part 1/3)**
>
> Thanks for your valuable comments and we hope our response fully resolves your concern.
>
> **[About the performance of Mega-TTS with 3-second prompts]**
> Thank you for raising this concern. We have evaluated the performance of Mega-TTS with 3-second prompts in the original version of the paper. In the experiment, for each sample, we randomly choose another utterance of the same speaker and crop a 3-second speech segment as the enrolled speech following VALL-E [1]. The results are shown in the following table. However, due to the limited page space, we removed it from the original paper. We agree that this result is important and have added it to the new version of the paper (marked in red).
>
> | Model     |  WER  |  SIM  |  DTW  |     QMOS      |     SMOS      |
> | --------- | :---: | :---: | :---: | :-----------: | :-----------: |
> | VALL-E-3s | 5.83% | 0.885 | 36.59 | 3.89$\pm$0.10 | 3.70$\pm$0.11 |
> | Ours-3s   | 2.46% | 0.898 | 34.39 | 3.99$\pm$0.06 | 3.75$\pm$0.10 |
>
>
>
> **[About the absence of comparisons with NaturalSpeech2 and VoiceBox]**
> We agree that the comparisons with NaturalSpeech 2 and VoiceBox are important for evaluating the proposed model's superiority. However, VALL-E [1], NaturalSpeech 2 [2], and Voice Box [3] have not released their codes or pre-trained models officially. As described in Appendix F, we have also tried to carefully reproduce NaturalSpeech 2 and VoiceBox but failed unfortunately. Here, we make comparisons with demo cases on the LibriSpeech test-clean set from their demo pages. To make fair comparisons, we use 3-second prompts for our Mega-TTS in this experiment. We only show the subjective metrics due to the limited number of demo cases. The results are shown in the following table. We can see that Mega-TTS outperforms VALL-E and VoiceBox and achieves a comparable performance with NaturalSpeech 2 when the length of the speech prompt is 3 seconds. Moreover, the performance can be further improved by the multi-sentence prompting mechanisms proposed in our paper.
>
> | Model    |       QMOS        |       SMOS        |
> | -------- | :---------------: | :---------------: |
> | VALL-E   |   3.93$\pm$0.12   |   3.79$\pm$0.10   |
> | Mega-TTS | **4.01$\pm$0.09** | **3.86$\pm$0.07** |
>
> | Model    |       QMOS        |       SMOS        |
> | -------- | :---------------: | :---------------: |
> | Voicebox |   3.84$\pm$0.09   |   3.75$\pm$0.11   |
> | Mega-TTS | **3.89$\pm$0.10** | **3.81$\pm$0.09** |
>
> | Model           |       QMOS        |     SMOS      |
> | --------------- | :---------------: | :-----------: |
> | NaturalSpeech 2 |   4.05$\pm$0.12   | 3.97$\pm$0.09 |
> | Mega-TTS        | **4.07$\pm$0.09** | 3.96$\pm$0.11 |
>
> **[About the assumption (1) ]**
> Yes, we agree that since we use the audiobook-style training corpus, the **global** speaking style across different utterances by the same speaker remains unchanged. Under this circumstance, the timbre encoder can capture **global** style information. On the other hand, the **local prosodic patterns** vary dramatically across different utterances by the same speaker. The P-LLM in our paper mainly aims at capturing the **fine-grained (local) prosodic patterns**.  To enhance the clarity of our paper, we have provided additional clarification for assumption (1) and explained that we only disentangle the fine-grained prosody and timbre information in the new version of the paper (marked in red). Furthermore, our experiments demonstrate the capability of Mega-TTS to separately control prosody and timbre in a zero-shot setting. Even in some emotional datasets (the ESD dataset) and in some in-the-wild corpus (as shown in the demo page), our method is also able to clone non-audiobook style voices effectively.
>
>
>
> **[About the detailed descriptions of the information bottleneck]**
> Thank you for raising this concern. Intuitively, there are four crucial variables for ensuring an appropriate information bottleneck: 1) the number of codebook embedding; 2) the codebook channel size; 3) the compression rate $r$ of the VQ Encoder; 4) and the downsampling rate $d$ of the MRTE. However, the search space of these hyperparameters is too large. Since the settings of r and d also influence the reconstruction quality, the burden of P-LLM, and the computational requirements, we first consider $r$ and $d$, fix them, and find the best setting for the hyperparameters of the codebook. We have introduced the detailed guidelines for setting the information bottleneck and designed experiments for evaluating the disentanglement performance in Appendix F in the new version of the paper (marked in red). Moreover, we also include the results with different sizes of training data to demonstrate that our system performs well with different data volumes consistently.

---

> ### Author Response · Authors · 2023-11-19
> **Response to Reviewer 6vRj (Part 2/3)**
>
> **[About using Montreal Forced Aligner (MFA)]**
> Yes, Mega-TTS needs an external aligner (Montreal Forced Aligner) to obtain the alignment beforehand. We agree that the data preprocessing pipeline is more complicated than those with end-to-end aligners. However, in our experiments, we find that the external aligner based methods are more robust in large-scale and multi-speaker scenarios. Here we conduct WER evaluations for models based on the external aligner and models based on end-to-end aligners (Like YourTTS [4] and VALL-E [1]). All systems are trained on the large-scale LibriLight dataset and tested on the LibriSpeech test-clean set, which is a zero-shot test setting. For fair comparisons, we keep the number of parameters consistent among all systems. The results are shown in the following table. We can see that the WER of the models based on the external aligner is significantly lower than those based on the end-to-end aligner, demonstrating their effectiveness in practical scenarios.
>
> |    Model     |                    Description                    |   Aligner Type   |    WER    |
> | :----------: | :-----------------------------------------------: | :--------------: | :-------: |
> |   YourTTS    |                Official Checkpoint                |    End-to-End    |   7.10%   |
> |   YourTTS    | Trained on LibriLight (1k hours), 400M parameters |    End-to-End    |   6.83%   |
> |    VALL-E    |         Trained on LibriLight (1k hours)          |    End-to-End    |  15.73%   |
> | FastSpeech 2 |         Trained on LibriLight (1k hours)          | External Aligner | **4.81%** |
> |   Mega-TTS   |         Trained on LibriLight (1k hours)          | External Aligner | **4.69%** |
>
>
>
> **[About the performance of using only the prosody information from another speaker]**
> Here we compare the use of prosody information solely from another speaker with the use of the interpolation technique. The results are presented in the table below. It is evident that relying only on prosody information from another speaker significantly increases the word error rate (WER). This phenomenon may be due to the fact that speakers A and B may not have identical vocal ranges. Directly transferring B's prosody to A may cause speech intelligibility issues, especially for some special prosody patterns. Our proposed prosody interpolation technique resolves this issue by transferring the prosody patterns in a soft and fine-grained manner during the autoregressive generation process.
>
> | Setting                             |    WER    |   SIM-AB    |    DE    |     QMOS      |  SMOS-AB  |
> | ----------------------------------- | :-------: | :---------: | :------: | :-----------: | :-------: |
> | Interpolation ($\gamma=0.8$)        | **4.82%** | 0.920/0.513 |   32.8   | 3.92$\pm$0.08 | 3.87/2.64 |
> | Another Speaker Only ($\gamma=1.0$) |   5.29%   | 0.918/0.510 | **32.2** | 3.84$\pm$0.12 | 3.88/2.60 |

---

> > ### Comment · Reviewer_6vRj · 2023-11-22
> > **Thanks for author's response**
> >
> > I thank the authors for the detailed response. My concerns are addressed well. I will raise my score from 5 to 6.

---

> ### Author Response · Authors · 2023-11-19
> **Response to Reviewer 6vRj (Part 3/3)**
>
> **[About the dimension in which the concatenation is performed in the Multi-Reference Timbre Encoder]**
> The concatenation is performed in the **time dimension** in the Multi-Reference Timbre Encoder. We are sorry that the corresponding part in Figure 1 is misleading. We have added more descriptions to the title of Figure 1in the rebuttal version of the paper (marked in red).
>
> **[About using a different model to measure SIM instead of using WavLM the one utilized by VALL-E [1]]**
> Yes, we agree that replicating the experimental conditions described in the baseline papers in the evaluation is important. Therefore, we use the WavLM utilized by VALL-E [1] to measure SIM and the results are shown in the following table. It can be seen that the trend and conclusion of the new experimental results are consistent with the results of the speaker verification model used in our paper.
> We acknowledge the significance of replicating the experimental conditions outlined in the baseline papers during the evaluation. As a result, we employ the WavLM, which was utilized by VALL-E [1], to assess SIM. The corresponding results are presented in the table below. Moreover, it is evident that the recent experimental findings align with the outcomes of the speaker verification model employed in our study.
>
> | Model         |    SIM    |
> | ------------- | :-------: |
> | GT            |   0.683   |
> | Baseline-10s  |   0.544   |
> | Baseline-60s  |   0.577   |
> | Baseline-300s | **0.591** |
> | VALL-E-3s     |   0.517   |
> | VALL-E-10s    |   0.539   |
> | VALL-E-20s    |   0.364   |
> | Ours-3s       | **0.551** |
> | Ours-10s      | **0.556** |
> | Ours-60s      | **0.582** |
> | Ours-300s     | **0.587** |
>
> **[References]**
> *[1] Wang, Chengyi, et al. "Neural codec language models are zero-shot text to speech synthesizers." arXiv preprint arXiv:2301.02111 (2023).*
> *[2] Shen, Kai, et al. "Naturalspeech 2: Latent diffusion models are natural and zero-shot speech and singing synthesizers." arXiv preprint arXiv:2304.09116 (2023).*
> *[3] Le, Matthew, et al. "Voicebox: Text-guided multilingual universal speech generation at scale." arXiv preprint arXiv:2306.15687 (2023).*
> *[4] Casanova, Edresson, et al. "Yourtts: Towards zero-shot multi-speaker tts and zero-shot voice conversion for everyone." International Conference on Machine Learning. PMLR, 2022.*

---

### Official Review · Reviewer_pM8F · 2023-11-04

**Soundness:** 3 good
**Presentation:** 3 good
**Contribution:** 3 good
**Rating:** 8
**Confidence:** 3

**Summary:**

The paper presents Mega-TTS, a novel framework for zero-shot text-to-speech (TTS) systems. The primary aim of Mega-TTS is to synthesize voices with unseen speech prompts, thereby reducing the data and computational requirements associated with voice cloning. The authors address two main challenges faced by existing zero-shot TTS systems: the lack of multi-sentence prompting strategies and the absence of specialized prompting mechanisms for prosodic information. By decomposing speech into content, timbre, and prosody, they propose a system that effectively handles long prompts and offers flexible control over prosodic styles. Experimental results suggest that Mega-TTS outperforms other state-of-the-art models in terms of speaker similarity and speech naturalness.

**Strengths:**

- The idea seems technically solid and well-motivated, and the demo audio examples clearly show the difference.

- The authors introduce a novel approach to decompose speech into content, timbre, and prosody. This method allows for more effective handling of long prompts and provides greater control over prosodic styles. This is an innovative contribution that sets the groundwork for future research in this area.

- Superior Performance: The paper presents experimental results showing that Mega-TTS outperforms other state-of-the-art models regarding speaker similarity and speech naturalness. This is a significant strength as it demonstrates the practical effectiveness of the proposed method.

**Weaknesses:**

Unclear Performance Across Languages: The experiments presented in the paper only use English datasets. Therefore, it's unclear how well the system performs with different languages or dialects. This limits the generalizability of the findings and may hinder the application of the system in diverse linguistic contexts. While the authors acknowledge some limitations of their approach, a more extensive exploration and testing of these constraints could have provided a more comprehensive understanding of the model. This additional analysis could guide future research addressing these limitations and further refining the Mega-TTS model.

The authors have not provided any information about the inference times of Mega-TTS compared to other models. This omission makes it difficult to evaluate the model's performance in real-world scenarios where speed may be as important as accuracy.

**Questions:**

- How does Mega-TTS handle non-English languages or different dialects? Exploring this could help assess the generalizability of the model across various linguistic contexts.
- What are the specific computational requirements of Mega-TTS compared to other models? This information is crucial for understanding the trade-offs involved in using Mega-TTS.
- How does the performance of Mega-TTS scale with the size of the training data? Understanding this can provide insights into how well the model might perform in scenarios with varying amounts of available data.
- How does Mega-TTS handle non-standard speech patterns such as shouting, laughing, or other forms of emotional expression? This question could illuminate the model's ability to accurately capture and reproduce a wider range of human speech nuances.

---

> ### Author Response · Authors · 2023-11-19
> **Response to Reviewer pM8F**
>
> We are grateful for your positive review and valuable feedback, and we hope our response fully resolves your concern.
>
> **[About the performance with different languages] (Question 1)**
> Here, we train our Mega-TTS model on the multilingual *WenetSpeech dataset (Chinese) + LibriLight dataset (English)* and use the *AISHELL-3 dataset (Chinese)* as the test set. We split the dataset and make evaluations according to Section 4.2 in our paper. To evaluate the WER for the Chinese dataset, we fine-tune the wav2vec2-large model. The results are shown in the following table. It can be seen that our Mega-TTS demonstrates superior performance with Chinese datasets. Due to the limited time constraints, we only conduct evaluations on the Chinese test set. Chinese audio examples and cross-lingual examples can be found in the "Additional Examples for Rebuttal" Section of our demo page: [https://boostprompt.github.io/boostprompt/](https://boostprompt.github.io/boostprompt/).
>
> | Dataset Usage |    WER    |    SIM    |    DTW    |       QMOS        |       SMOS        |
> | ------------- | :-------: | :-------: | :-------: | :---------------: | :---------------: |
> | GT            |   2.26%   |     -     |     -     |   4.02$\pm$0.07   |   4.23$\pm$0.09   |
> | Baseline-10s  |   5.76%   |   0.906   |   21.43   |   3.87$\pm$0.09   |   4.01$\pm$0.08   |
> | VALL-E-10s    |   7.01%   |   0.901   |   19.52   |   3.80$\pm$0.13   |   3.98$\pm$0.08   |
> | Ours-10s      | **3.13%** | **0.914** | **18.40** | **3.92$\pm$0.10** | **4.05$\pm$0.08** |
>
> **[About the inference times and computational requirements of Mega-TTS compared to other models] (Question 2)**
> We agree that the computational requirements are crucial for understanding the trade-offs involved in using Mega-TTS. In Table 1, we have provided the real-time factor (RTF),  the time (in seconds) required for the system to synthesize one-second waveform, in the inference stage of Mega-TTS compared to other models. Besides, we have also compared the total number of parameters in Table 1 and provided the computational requirements for training Mega-TTS in Section 4.1.
>
> **[About the performance of Mega-TTS scale with different sizes of the training data]**
> Here we evaluated the performance of our Mega-TTS scale with varying amounts of available data. In this experiment, all of the systems use 3 seconds of speech prompts. The results are shown in the following table. We can see that Mega-TTS performs well with different sizes of the training data, while VALL-E fails to obtain satisfying results when the data is insufficient. We also scale our Mega-TTS with 200K hours of speeches and the results can be found in Appendix B in the original version of the paper.
>
> | Model    |     Dataset Usage      |     WER     |     SIM     |     DTW     |        QMOS         |        SMOS         |
> | -------- | :--------------------: | :---------: | :---------: | :---------: | :-----------------: | :-----------------: |
> | VALL-E   | LibriLight (1k hours)  |   15.73%    |    0.782    |    54.05    |    3.63$\pm$0.13    |    3.52$\pm$0.10    |
> | VALL-E   | LibriLight (10k hours) |    8.29%    |    0.870    |    38.93    |    3.84$\pm$0.12    |    3.65$\pm$0.12    |
> | VALL-E   | LibriLight (60k hours) |    5.83%    |    0.885    |    36.59    |    3.90$\pm$0.12    |    3.71$\pm$0.10    |
> | Mega-TTS | LibriLight (1k hours)  |    4.69%    |    0.861    |    43.07    |    3.88$\pm$0.11    |    3.61$\pm$0.10    |
> | Mega-TTS | LibriLight (10k hours) |    2.91%    |    0.882    |    35.76    |    3.95$\pm$0.11    |    3.70$\pm$0.12    |
> | Mega-TTS | LibriLight (60k hours) | ***2.46%*** | ***0.898*** | ***34.39*** | ***3.98$\pm$0.09*** | ***3.77$\pm$0.08*** |
>
> **[About handling non-standard speech patterns]**
> We have generated some speech samples by directly utilizing non-standard speech voice prompts, such as laughing and other forms of emotional expression. The generated speech samples can be accessed in the "Additional Examples for Rebuttal" Section of our demo page: [https://boostprompt.github.io/boostprompt/](https://boostprompt.github.io/boostprompt/). From the demo page, we can see that Mega-TTS can accurately capture timbre information even from a short clip of a non-standard speech pattern. This demonstrates its effectiveness in "capturing and reproducing a wider range of human speech nuances". Again, we greatly appreciate your valuable suggestions and assistance.

---

> ### Comment · Area_Chair_cyXG · 2023-11-22
> **Reminder to respond to authors' rebuttal**
>
> Dear Reviewer,
>
> Please respond to authors rebuttal and see whether they have addressed your concerns. Thanks!

---

### Author Response · Authors · 2023-11-19
**Summary of the rebuttal revision**

We would like to thank the reviewers for their constructive reviews! Here we summarize the main revision of the manuscript according to the comments and suggestions of reviewers:

- In Section 2, we include detailed discussions on the attention-based adaptation methods and highlight the main differences between them and our own approach.
- In Section 3.1, we provide additional clarification for assumption (1) and explain that we only disentangle the fine-grained prosody and timbre information to enhance the clarity of our paper
- In Table 1, we add the results of "ours-3s" to make a fairer comparison with VALL-E.
- In Appendix F, we add detailed guidelines for setting variables such as $r$, $d$, and hidden dimensions to ensure an appropriate information bottleneck.
- In Appendix G, we evaluate the performance of our Mega-TTS scale with varying amounts of available data.
- In Appendix H, we add the ablation studies about the word-level, phoneme-level, and stride-8-level prosody modeling strategies.
- In Appendix J, we make ablation studies for different lengths of context during Mega-TTS’s training process to further verify the effectiveness of the proposed multi-sentence prompting mechanisms.
- In Appendix L, we conduct experiments on LibriSpeech test-other set to verify our model’s robustness against noisy reference prompts.

The audio samples of these additional experiments can also be found in the "Additional Examples for Rebuttal" Section of our demo page: [https://boostprompt.github.io/boostprompt/](https://boostprompt.github.io/boostprompt/). We greatly appreciate the reviewers' great efforts and valuable comments, which have significantly improved the soundness of our manuscript. We hope the provided responses adequately address the issues raised. As the end of the rebuttal phase is approaching, we would be grateful if we could hear your feedback regarding our answers to the reviews. We will be very happy to clarify any remaining points (if any).

---

### Author Response · Authors · 2023-11-22

We thank all the reviewers for taking the time to review our paper. We have addressed the comments from each reviewer. We would like to respectfully remind everyone that the author-reviewer discussion period will soon conclude. We would greatly appreciate any further feedback on our rebuttal.

---

### Meta-Review · Area_Chair_cyXG · 2023-12-06

**Metareview:**

The paper presents a zero-shot TTS system - Mega-TTS. It decomposes speech into content, timbre and prosody. With this decomposition, the system can effectively handles long prompts and offers flexible control over prosodic styles. Experimental validations have demonstrated that Mega-TTS outperforms other SOTA systems.

Strengths:
The proposed approach to decompose speech into content, timbre, and prosody is interesting. This brings the capability to independently prompt prosody and timbre within a zero-shot setting. It also enables scaling the in-context learning to very long prompts.

Weaknesses:
As pointed by the reviewers, there was some concerns on baselines, datasets, multilingual and noisy conditions. The authors have addressed them with additional experiments. However, some baselines are not successfully reproduced and limited samples from demos were used for comparisons.

**Justification For Why Not Higher Score:**

The proposed Mega-TTS demonstrated good zero-shot TTS capabilities. The decomposition of speech to content, timbre and prosody is neat but novelty wise it's not ground breaking findings.

**Justification For Why Not Lower Score:**

Interesting study with good quality gains.

---

### Decision · Program_Chairs · 2024-01-16

Accept (poster)